# The GPR120 agonist TUG-891 promotes metabolic health by stimulating mitochondrial respiration in brown fat

Maaike Schilperoort[1,2,3,*] iD, Andrea D van Dam[2,3], Geerte Hoeke[2,3], Irina G Shabalina[4] iD, Anthony Okolo[5], Aylin C Hanyaloglu[5], Lea H Dib[6], Isabel M Mol[2,3], Natarin Caengprasath[5], Yi-Wah Chan[7], Sami Damak[8], Anne Reifel Miller[9], Tamer Coskun[9], Bharat Shimpukade[10], Trond Ulven[10], Sander Kooijman[2,3], Patrick CN Rensen[2,3] & Mark Christian[1,**] iD

## Abstract

Brown adipose tissue (BAT) activation stimulates energy expenditure in human adults, which makes it an attractive target to combat obesity and related disorders. Recent studies demonstrated a role for G protein-coupled receptor 120 (GPR120) in BAT thermogenesis. Here, we investigated the therapeutic potential of GPR120 agonism and addressed GPR120-mediated signaling in BAT. We found that activation of GPR120 by the selective agonist TUG-891 acutely increases fat oxidation and reduces body weight and fat mass in C57Bl/6J mice. These effects coincided with decreased brown adipocyte lipid content and increased nutrient uptake by BAT, confirming increased BAT activity. Consistent with these observations, GPR120 deficiency reduced expression of genes involved in nutrient handling in BAT. Stimulation of brown adipocytes *in vitro* with TUG-891 acutely induced $O_2$ consumption, through GPR120-dependent and GPR120-independent mechanisms. TUG-891 not only stimulated GPR120 signaling resulting in intracellular calcium release, mitochondrial depolarization, and mitochondrial fission, but also activated UCP1. Collectively, these data suggest that activation of brown adipocytes with the GPR120 agonist TUG-891 is a promising strategy to increase lipid combustion and reduce obesity.

**Keywords** brown adipose tissue; $Ca^{2+}$; GPR120; lipid metabolism; mitochondria
**Subject Categories** Metabolism; Pharmacology & Drug Discovery

## Introduction

Brown adipose tissue (BAT) is present and active in human adults and contributes to total energy expenditure (EE) (Cypess *et al*, 2009; van Marken Lichtenbelt *et al*, 2009; Virtanen *et al*, 2009). This contrasts with white adipose tissue (WAT), which primarily serves as a site of energy storage. Cold exposure, the natural stimulus of BAT, increases the volume and activity of metabolically active BAT and reduces fat mass in adult human subjects (van der Lans *et al*, 2013; Yoneshiro *et al*, 2013; Blondin *et al*, 2014). Cold exposure stimulates the sympathetic nervous system to release norepinephrine, which in turn activates brown adipocytes through the β3-adrenergic receptor (ADRB3) (Argyropoulos & Harper, 2002). Activation of brown adipocytes initiates intracellular signaling cascades, resulting in the breakdown of triglycerides (TG) stored in intracellular lipid droplets to yield fatty acids (FA) and glycerol (Cannon & Nedergaard, 2004). The FAs are subsequently transported to the mitochondria where they are either oxidized or used to allosterically activate uncoupling protein-1 (UCP1), which is present in the inner membrane of mitochondria (Fedorenko *et al*, 2012; Nicholls, 2017). UCP1 disrupts the proton gradient that is required for ATP synthesis, resulting in the release of energy as heat

1  Division of Biomedical Sciences, Warwick Medical School, University of Warwick, Coventry, UK
2  Division of Endocrinology, Department of Medicine, Leiden University Medical Center, Leiden, The Netherlands
3  Einthoven Laboratory for Experimental Vascular Medicine, Leiden, The Netherlands
4  Department of Molecular Biosciences, The Wenner-Gren Institute, The Arrhenius Laboratories F3, Stockholm University, Stockholm, Sweden
5  Department of Surgery and Cancer, Institute of Reproductive and Developmental Biology, Imperial College London, London, UK
6  Institute of Cardiovascular Sciences, College of Medical and Dental Sciences, University of Birmingham, Birmingham, UK
7  Lymphocyte Development Group, MRC London Institute of Medical Sciences, Hammersmith Campus, Imperial College London, London, UK
8  Nestlé Research Center, Lausanne, Switzerland
9  Lilly Research Laboratories, Diabetes/Endocrine Department, Lilly Corporate Center, Indianapolis, IN, USA
10  Department of Physics, Chemistry and Pharmacy, University of Southern Denmark, Odense, Denmark
   *Corresponding author. Tel: +31-71-5265304. E-mail: m.schilperoort@lumc.nl
   **Corresponding author. Tel: +44-24-76-968585. E-mail: m.christian@warwick.ac.uk

instead of ATP: a process called thermogenesis (Trayhurn, 2017). Since activated BAT burns high amounts of FAs, it is considered an attractive target to combat obesity and related disorders. Therefore, novel targets to increase BAT activity are highly warranted.

A potential target is G protein-coupled receptor 120 (GPR120), also termed free FA receptor 4 (FFAR4). We have previously shown that GPR120 is highly expressed in BAT and cold exposure further increases its expression in both BAT and subcutaneous WAT of mice (Rosell *et al*, 2014), suggesting that GPR120 contributes to the thermogenic capacity of BAT. GPR120 is activated by both medium-chain FA (MCFA) and long-chain FA (LCFAs) (Hirasawa *et al*, 2005; Christiansen *et al*, 2015) and is coupled to Gαq, which activates several intracellular signaling pathways. Recent studies have revealed that through these signaling mechanisms, GPR120 plays an important role in energy metabolism, hormonal regulation, and the immune system. For example, Oh *et al* (2010) demonstrate that GPR120 mediates the anti-inflammatory actions of ω-3 FAs. GPR120 deficiency leads to obesity, glucose intolerance, and hepatic steatosis in mice fed a high-fat diet (Ichimura *et al*, 2012). In humans, *GPR120* expression is higher in obese compared to lean subjects, and individuals carrying a mutation associated with decreased GPR120 signaling have an increased risk of obesity (Ichimura *et al*, 2012). Given the high GPR120 expression in BAT, it is likely that BAT contributes to the metabolic effects of GPR120 observed in these studies. Indeed, a very recent study by Quesada-López *et al* (2016) confirmed a role for GPR120 in BAT activation. However, therapeutic potential and underlying signaling of GPR120-mediated BAT activation remain to be elucidated.

Therefore, the aims of this study were to further investigate the therapeutic potential of GPR120 agonism and to address GPR120-mediated intracellular signaling in BAT. We found that stimulation of GPR120 by the agonist TUG-891 increases fat oxidation and lipid uptake by BAT thereby reducing fat mass, while GPR120 deficiency reduces expression of genes involved in nutrient handling. Mechanistically, we show that TUG-891 acts in a GPR120-dependent manner to induce intracellular $Ca^{2+}$ release which could result in mitochondrial depolarization and fragmentation. In addition, our data reveal that TUG-891 activates mitochondrial UCP1, which may act synergistically with mitochondrial fragmentation to increase respiration. Taken together, our data indicate that by acutely increasing lipid combustion by BAT, GPR120 agonism may be a promising therapeutic strategy to reduce obesity.

## Results

### The GPR120 agonist TUG-891 increases lipid oxidation and reduces fat mass in mice

To investigate the effect of GPR120 activation on energy metabolism *in vivo*, mice were injected with the GPR120 agonist TUG-891 daily for a period of 2.5 weeks. This compound was selected due to higher selectivity for GPR120 over GPR40 compared to other agonists, including GW9508 and NCG21 (Shimpukade *et al*, 2012; Hudson *et al*, 2013). TUG-891 reduced total body weight (Fig 1A), which was due to a large reduction in fat mass (−73%; Fig 1B) and a minor reduction in lean mass (−9.9%; Fig 1C) at week 2.5 compared to vehicle. The reduced lean mass could be due to increased muscle turnover, as TUG-891 non-significantly increased expression of markers for both muscle atrophy and regeneration

(Appendix Fig S1). During the first week of treatment, food intake was similar in the control and treatment groups (Fig 1D), while fat mass was already reduced by 19% in the TUG-891-treated group at day 5. Longer treatment reduced food intake, which further contributed to body weight and fat mass loss. To investigate whether TUG-891 enhances EE or alters substrate utilization, mice were housed in metabolic cages during the first week of treatment. TUG-891 treatment did not increase total EE (Appendix Fig S2A) nor did it affect physical activity levels (Appendix Fig S2B). However, TUG-891 acutely lowered the respiratory exchange ratio (RER) upon injection, which persisted throughout the dark period (Fig 1E). Accordingly, TUG-891 lowered glucose oxidation (Fig 1F) and largely increased fat oxidation (Fig 1G). This increase in fat oxidation was supported by histological analysis of adipose tissues, revealing that TUG-891 administration reduced lipid content in BAT (−28%; Fig 2A), and adipocyte size in both sWAT (−47%; Fig 2B) and gWAT (−38%; Fig 2C). In addition, total organ weights of iBAT (−31%), gWAT (−44%), and liver (−14%) were reduced in TUG-891-treated mice as compared to controls (Fig 2D). Plasma TG levels were increased at the end of the study, possibly as a result of increased lipolysis (Appendix Fig S3A). Protein (Appendix Fig S3B–E) and gene (Appendix Fig S3F) expressions of markers for lipolysis, adipogenesis, proliferation, and thermogenesis were largely unaffected in BAT. However, *Ucp1* gene expression (Appendix Fig S3H) and protein staining (Appendix Fig S4) were increased in gWAT of TUG-891-treated animals, suggesting GPR120-mediated browning.

As TUG-891 also has affinity for GPR40 (Hudson *et al*, 2013), we aimed to confirm that the effects of TUG-891 on body composition and substrate utilization were mediated by GPR120. To this end, metabolic effects of TUG-891 were also assessed in GPR120 KO mice and WT littermates. In GPR120 KO mice, TUG-891 non-significantly reduced body weight (Fig 3A) and fat mass (Fig 3B), but not to the same extent as in WT mice. TUG-891 decreased food intake similarly in WT and GPR120 KO mice (Fig 3C). The modest decrease in fat mass in TUG-891 GPR120 KO mice compared to non-treated WT mice may be related to diminished food intake. Lean mass was unchanged in all treatment groups (Fig 3D). In addition, while RER and fat oxidation did not differ between WT and GPR120 KO mice at baseline (Appendix Fig S5), TUG-891 non-significantly ($P = 0.136$) decreased RER (Fig 3E) and increased fat oxidation (Fig 3F) during the dark period in WT mice but not in GPR120 KO mice.

### The GPR120 agonist TUG-891 stimulates fatty acid uptake by BAT

Hereafter, we aimed to elucidate which organs were responsible for the increased fat oxidation in TUG-891-treated WT animals. As increased fat oxidation subsequently leads to increased FA uptake, the tissue-specific uptake of FAs derived from intravenously injected lipoprotein-like particles labeled with glycerol tri[³H]oleate was determined. In WT mice, TUG-891 treatment markedly increased the uptake of [³H]oleate in both iBAT and subscapular BAT (sBAT) as compared to vehicle (Fig 4A), suggesting increased BAT activity. TUG-891 also increased the uptake of [¹⁴C]deoxyglucose in iBAT and sBAT (Fig 4B). However, when the uptake data were corrected for organ weight (for organs that could be removed quantitatively within an acceptable time frame), the difference in glucose uptake was lost. FA uptake in whole iBAT remained approximately twice as high in treated WT mice versus controls (Appendix Fig S6),

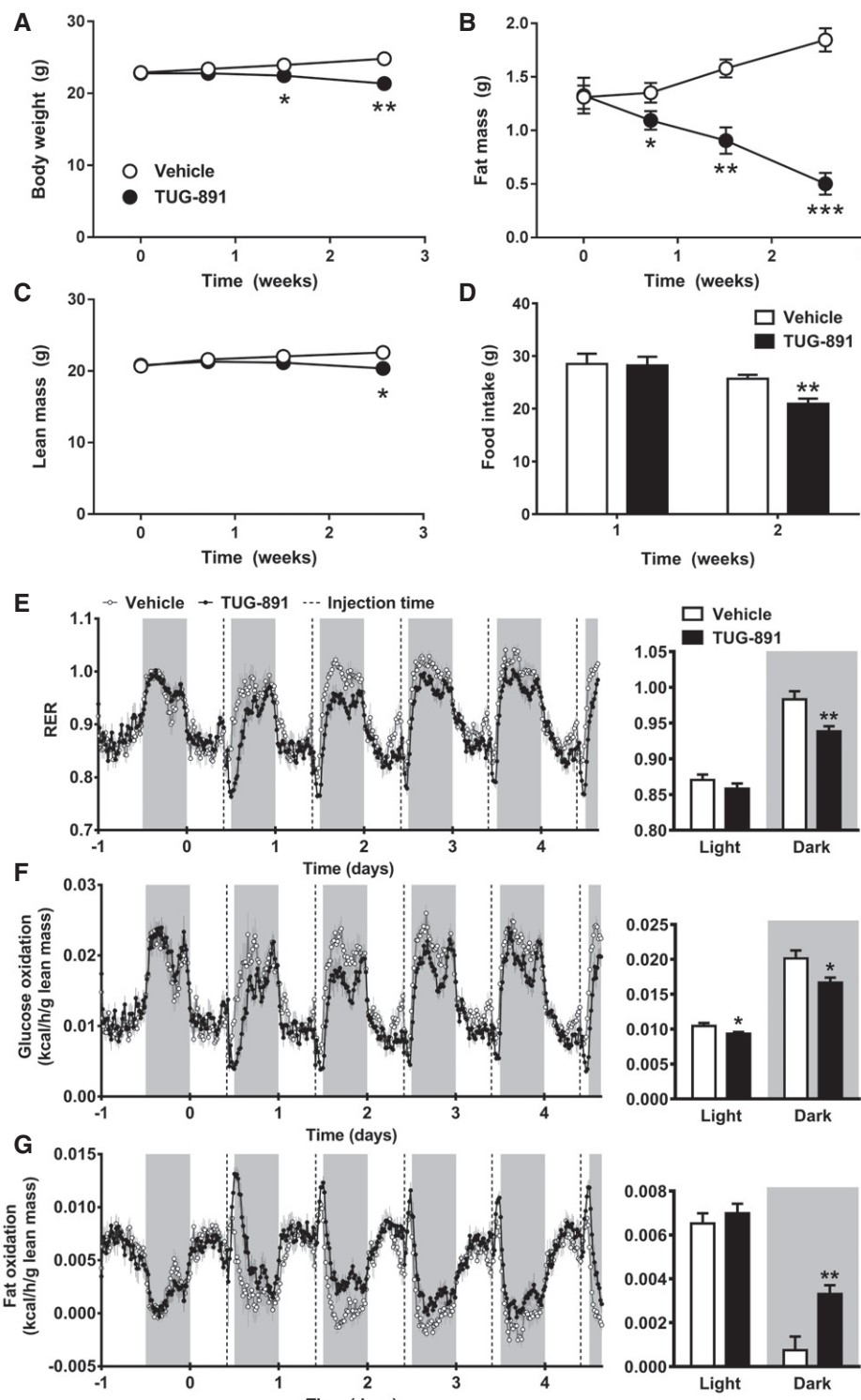

**Figure 1. The GPR120 agonist TUG-891 decreases body weight and fat mass, and increases fat oxidation.**

A–D   C57Bl/6J mice on chow diet were treated with the GPR120 agonist TUG-891 (35 mg/kg) or vehicle (*n* = 8) for 2.5 weeks. Body weight, fat mass, lean mass, and food intake were measured at indicated time points.

E–G   Vehicle- and TUG-891-treated mice (*n* = 8) were housed in fully automated metabolic cages in which respiratory exchange ratio (RER) (E), glucose oxidation (F), and fat oxidation (G) were measured. Injection of the GPR120 agonist TUG-891 (35 mg/kg) or vehicle is indicated by dotted lines, and light and gray areas represent the light and dark phase, respectively. For bar graph analysis, mean results in light and dark phase were calculated.

Data information: Data represent means ± SEM. *$P < 0.05$, **$P < 0.01$, ***$P < 0.001$ compared to the vehicle group, according to the two-tailed unpaired Student's *t*-test. The exact *P*-value for each significant difference can be found in Appendix Table S5.

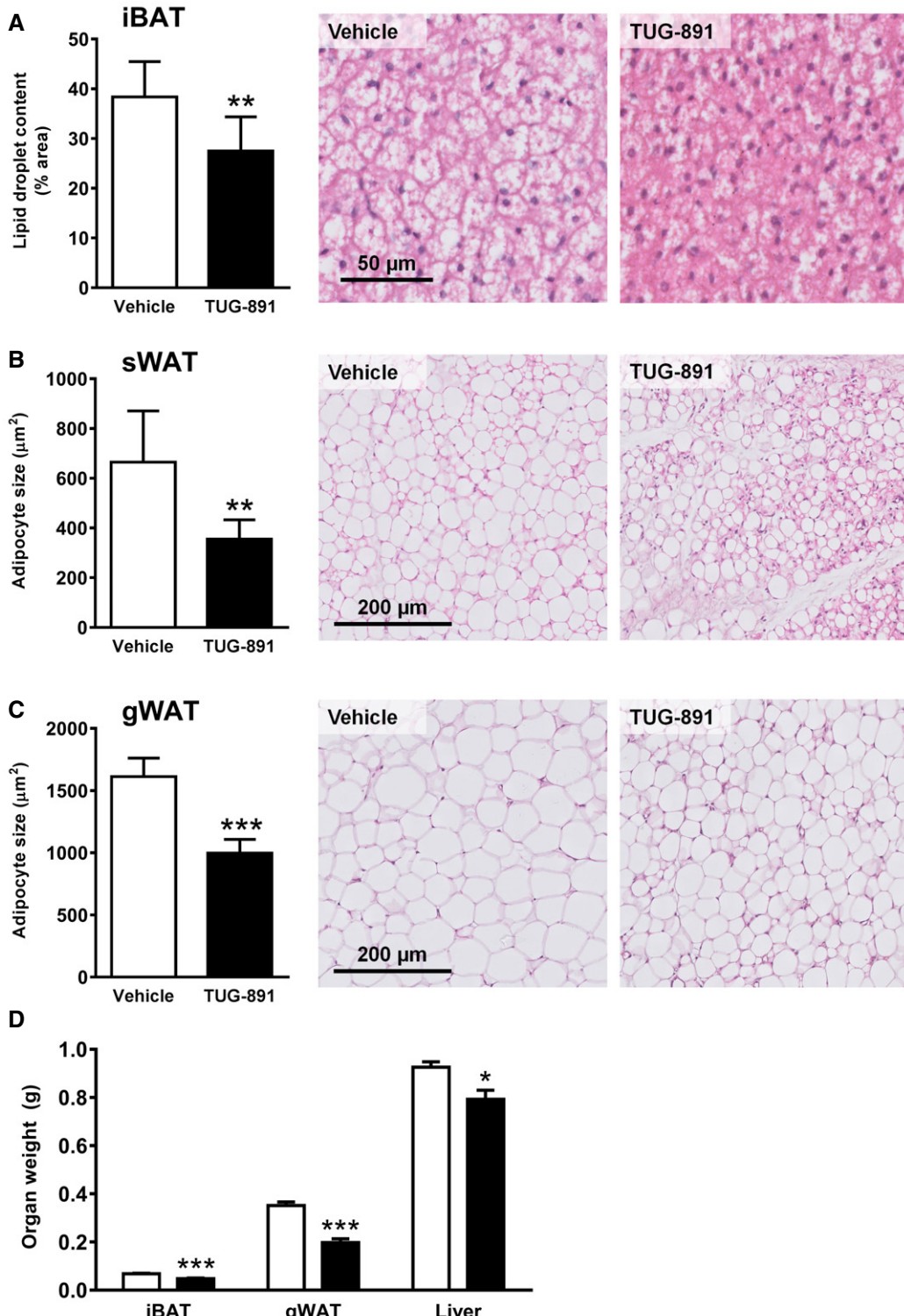

**Figure 2. TUG-891 decreases lipid content of BAT and WAT.**

A–C Representative images of hematoxylin and eosin (H&E)-stained interscapular BAT (iBAT), subcutaneous WAT (sWAT), and gonadal WAT (gWAT) of mice treated with vehicle or the GPR120 agonist TUG-891 ($n = 8$). Stained slides were digitalized, and lipid droplet content of BAT and adipocyte size in WAT was analyzed using ImageJ software.

D After mice treated with vehicle or the GPR120 agonist TUG-891 ($n = 8$) were sacrificed, iBAT, gWAT, and liver were collected and weighed ($n = 8$).

Data information: Data represent means $\pm$ SEM. *$P < 0.05$, **$P < 0.01$, ***$P < 0.001$ compared to the vehicle group, according to the two-tailed unpaired Student's $t$-test. The exact $P$-value for each significant difference can be found in Appendix Table S5.

 

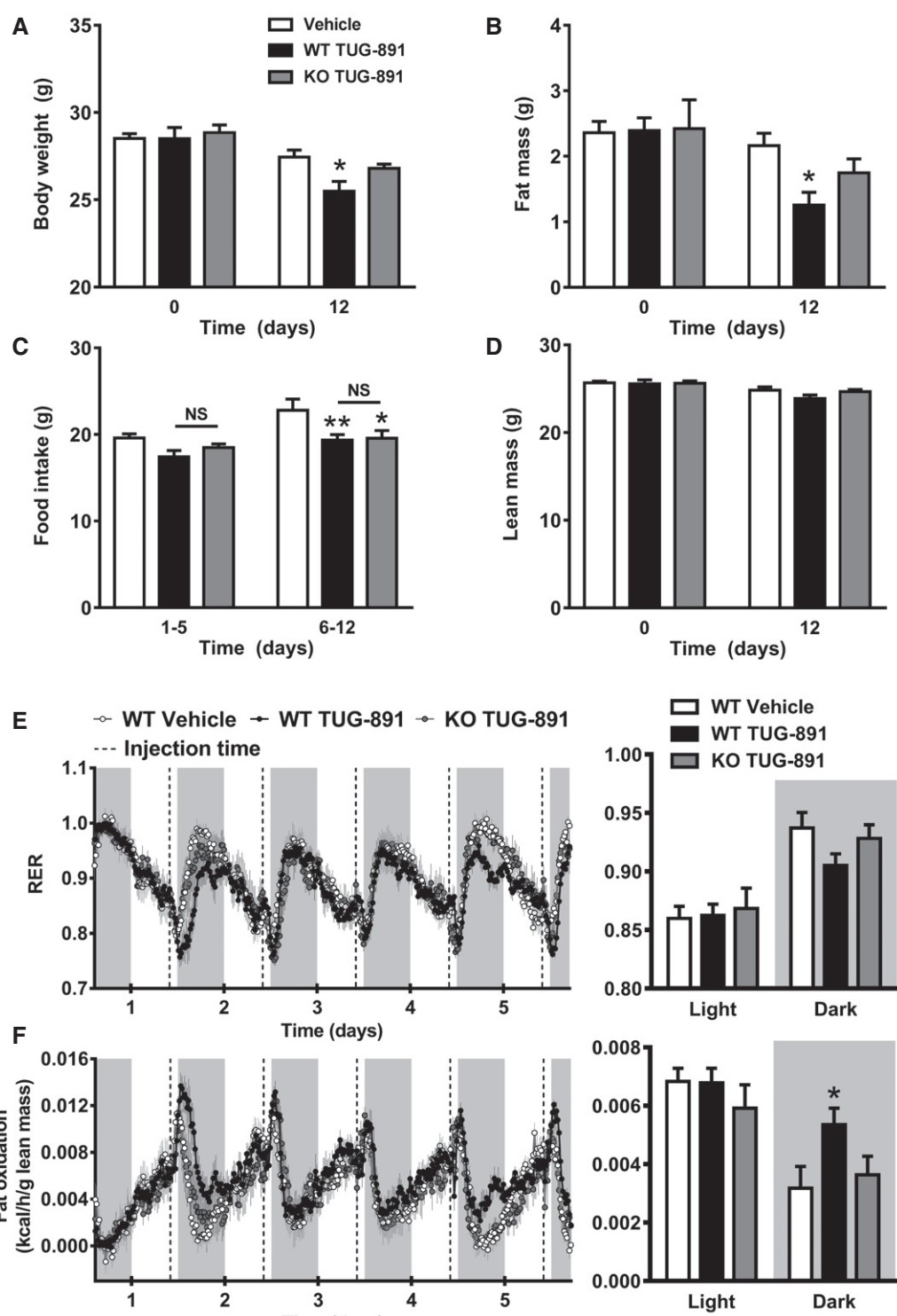

**Figure 3.  Metabolic effects of TUG-891 are reduced or absent in GPR120-deficient mice.**

A–D  GPR120 KO mice and WT littermates (*n* = 6–8) were treated with the GPR120 agonist TUG-891 (35 mg/kg) or vehicle for 12 days. At the beginning (day 0) and end (day 12) of this treatment period, body weight, fat mass, and lean mass were measured. Food intake was determined after 5 and 12 days of treatment.

E–F  GPR120 KO mice and WT littermates (*n* = 6–8) were treated with the GPR120 agonist TUG-891 (35 mg/kg) or vehicle. Respiratory exchange ratio (RER) and fat oxidation were determined by housing the mice in metabolic cages. Injection of TUG-891 or vehicle is indicated by dotted lines, and light and gray areas represent the light and dark phase, respectively. For bar graph analysis, mean results in the light and dark phase were calculated.

Data information: Data represent means ± SEM. *$P < 0.05$, **$P < 0.01$ compared to the vehicle group, according to two-way ANOVA with Tukey's *post hoc* test. The exact *P*-value for each significant difference can be found in Appendix Table S5.

showing an independency of organ weight. In line with these data, TUG-891 decreased total organ weights of iBAT and sWAT depots as compared to vehicle in WT mice, but not in GPR120 KO mice (Fig 4C).

## GPR120 alters expression of genes involved in nutrient handling

To evaluate how GPR120 modulates lipid handling by BAT, we investigated the effects of GPR120 deficiency on the global gene expression profile in BAT by performing a microarray on BAT of GPR120 KO and WT littermates. Clustering of genes was observed between GPR120 KO and WT mice (Fig 5A). The top 50 of genes that were either upregulated or downregulated in the absence of GPR120 are listed in Appendix Table S3. Selected genes were validated, and expression of genes associated with inflammation (Fig 5B), adipocyte biology (Fig 5C), glucose metabolism (Fig 5D), and lipid metabolism (Fig 5E) was investigated by qRT–PCR. Expression of inflammatory genes tended to be increased in GPR120 KO BAT. GPR120 deficiency upregulated *Sncg*, encoding synuclein-γ which is involved in lipid droplet dynamics in white adipocytes and is negatively regulated by PPARγ (Dunn *et al*, 2015). On the other hand, GPR120 deficiency downregulated *Mlxipl*, which encodes the carbohydrate response element-binding protein (ChREBP), a transcriptional inducer of glucose metabolism and *de novo* lipogenesis (Witte *et al*, 2015). Of the genes associated with glucose metabolism, *Glut4*, *Insr*, *Adcy4*, and *Gys2* were downregulated in GPR120 KO BAT. *Gys2* encodes glycogen synthase 2 and is PPARγ-regulated in adipocytes (Mandard *et al*, 2007). Of the genes that determine lipid metabolism, those involved in both lipogenesis (*Acc1*, *Acc2*, *Fas*, *Scd2*) and intracellular lipolysis (*Hsl*, *Atgl*, *Pnpla3*) were lower in GPR120 KO BAT.

Functional annotation clustering using DAVID (https://david.ncifcrf.gov/) (Dennis *et al*, 2003) revealed that genes downregulated in the absence of GPR120 were associated with mitochondrial function, FA metabolism, nucleotide binding, and mRNA processing (Appendix Table S4). The set of upregulated genes was associated with immune responses, as well as antigen processing and ribosomes.

## Gpr120 expression promotes brown adipocyte differentiation and is increased in "browned" white adipocytes

Using a conditionally immortalized model of brown adipocytes (Rosell *et al*, 2014), we investigated the expression profile of *Gpr120* in preadipocytes differentiated to fully mature adipocytes over 7 days. Like *Ucp1* expression, *Gpr120* expression was highly induced during differentiation of brown adipocytes, reaching maximum levels on day 6 (Fig 6A). This is consistent with high GPR120 expression in BAT compared to other organs (Appendix Fig S7). Treatment of differentiated adipocytes with the β3-adrenergic agonist CL induced both *Gpr120* (ninefold) and *Ucp1* (53-fold) expression (Fig 6A). Differentiation also increased expression of adipocyte markers *aP2*, *Cidea*, and *Adrb3*, and decreased expression of the preadipocytes marker *Pref1*, validating our brown adipocyte cell line (Appendix Fig S8).

As we previously reported that *Gpr120* was induced by cold exposure in white adipose tissue (Rosell *et al*, 2014), we next investigated whether *in vitro* browning of white adipocytes with

rosiglitazone treatment (resulting in so called "brite" adipocytes) could similarly enhance gene expression. Unlike *Ucp1* that was induced by CL in brown, white, and brite adipocytes, *Gpr120* expression was not increased by CL treatment in white and brite adipocytes (Fig 6B). However, basal expression of *Gpr120* in differentiated brite adipocytes was increased as compared to white adipocytes, indicating a potential role of *Gpr120* in browning of white adipocytes.

To study whether GPR120 is directly involved in adipocyte differentiation, brown adipocyte cell lines were generated from WT and GPR120 KO mice. Both cell lines differentiated to mature brown adipocytes when exposed to a standard hormone differentiation treatment. However, GPR120 KO adipocytes accumulated a lower amount of lipids as evidenced by Oil Red O staining (Fig 6C) and exhibited lower expression of the adipocyte differentiation marker *aP2* and *Ucp1* (Fig 6D), suggesting impaired differentiation in GPR120 KO cells. Treatment with TUG-891 throughout differentiation tended to increase *Ucp1* expression in WT but not GPR120 KO cells (Fig 6D).

## TUG-891 directly activates brown adipocytes *in vitro* through UCP1 activation and mitochondrial fragmentation

To investigate whether the GPR120 agonist TUG-891 directly activates brown adipocytes and to study the downstream intracellular signaling pathways involved, we stimulated differentiated brown adipocytes with TUG-891. Strikingly, TUG-891 acutely increased the $O_2$ consumption rate (OCR) of brown adipocytes by more than twofold (Fig 7A). Pretreatment with the GPR120 antagonist AH7614 reduced rather than abolished this response (Fig 7A), indicating that TUG-891 exhibits both GPR120-dependent and GPR120-independent activity. We investigated whether TUG-891 functions in a manner similar to LCFAs which can directly activate UCP1 by measuring $O_2$ consumption in isolated BAT mitochondria in conditions mimicking a cellular environment with high purine nucleotide (GDP) content and inhibited UCP1 (Matthias *et al*, 2000). Indeed, TUG-891 ($\geq 10\ \mu M$) increased $O_2$ consumption in mitochondria isolated from WT mice (Fig 7B), suggesting that TUG-891 has the capacity to overcome purine nucleotide inhibition and activate UCP1 in brown adipocytes. TUG-891 also increased $O_2$ consumption in mitochondria from UCP1 KO mice, but this effect was smaller and occurred at higher concentrations ($\geq 90\ \mu M$) as compared to WT mitochondria (Fig 7C), a response that is also observed with oleate (Shabalina *et al*, 2004). As oxidative capacity (FCCP response, Fig 7B and C) of WT and UCP1 KO mitochondria was equal, these results suggest that TUG-891 increases mitochondrial respiration through activation of UCP1 (Appendix Fig S9A). TUG-891 exhibited a competitive interaction with GDP in WT but not UCP1 KO mitochondria (Appendix Fig S9B and C), further supporting the effect of TUG-891 on UCP1.

Next, we investigated GPR120-dependent effects of TUG-891 by examining potential downstream targets of G protein signaling that could partly mediate the TUG-891-induced $O_2$ consumption in brown adipocytes. To ensure that GPR120 is Gαq-coupled and does not signal via Gαs in brown adipocytes, the effect of TUG-891 on intracellular cAMP levels was determined. As expected, TUG-891 had no effect on cAMP production (Appendix Fig S9D). As for Gαq targets, TUG-891 increased the amount of phosphorylated ERK and AKT (Appendix Fig S9E). However, pretreatment of brown

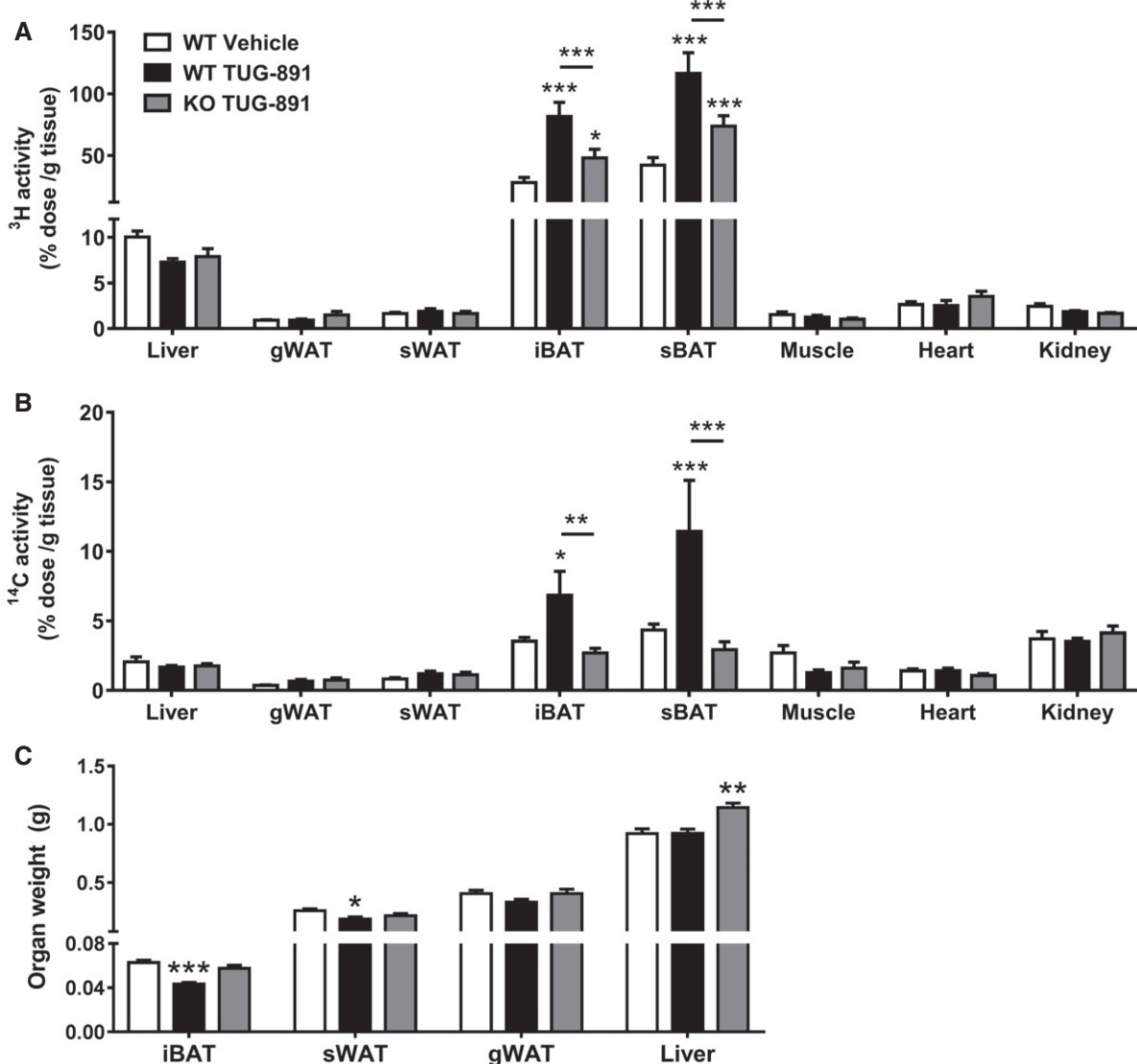

**Figure 4.  TUG-891 increases the uptake of nutrients by BAT.**

A, B    WT and GPR120 KO mice (*n* = 6–8) treated with vehicle or the GPR120 agonist TUG-891 were intravenously injected with [³H]TO-labeled lipoprotein-like emulsion particles and [¹⁴C]deoxyglucose ([¹⁴C]DG). After 15 min, mice were sacrificed and uptake of [³H]TO- and [¹⁴C]DG-derived radioactivity per gram tissue was determined in various organs, including gonadal WAT (gWAT), subcutaneous WAT (sWAT), interscapular BAT (iBAT), and subscapular BAT (sBAT).

C    After WT and GPR120 KO mice (*n* = 6–8) treated with vehicle or the GPR120 agonist TUG-891 were sacrificed, organs were collected and weighed.

Data information: Data represent means ± SEM. *$P < 0.05$, **$P < 0.01$, ***$P < 0.001$ compared to the vehicle group or indicated control group, according to two-way ANOVA with Tukey's *post hoc* test (A, B) or the two-tailed unpaired Student's *t*-test (C). The exact *P*-value for each significant difference can be found in Appendix Table S5.

adipocytes with the MEK inhibitor U0126 (Appendix Fig S9F) or an AKT 1/2 kinase inhibitor (Appendix Fig S9G) did not reduce $O_2$ consumption, excluding requirement of the ERK and AKT pathways for this effect. Pretreatment with the cell-permeable $Ca^{2+}$ chelator BAPTA-AM strongly reduced the TUG-891-induced $O_2$ consumption (Fig 7D), indicating that intracellular $Ca^{2+}$ is essential for GPR120-mediated activation of brown adipocytes. Indeed, TUG-891 strongly increased intracellular $Ca^{2+}$ concentrations (Fig 7E). This effect was absent in adipocytes preincubated with the GPR120 antagonist AH7614 and in GPR120 KO adipocytes (Fig 7E), confirming GPR120

dependency. The $G\alpha q$ inhibitor YM-254890 also blocked the $Ca^{2+}$ response (Appendix Fig S9H), indicating that this effect of GPR120 activation is indeed mediated via $G\alpha q$ signaling. As $Ca^{2+}$ could affect mitochondrial polarization, effects of TUG-891 on mitochondrial membrane potential were investigated. Cells were incubated with MitoTracker Green FM (MTG) and MitoTracker Red CMXRos (MTR), which stain mitochondria independent of and dependent on membrane potential, respectively. Relative intensity (MTR/MTG) of these stainings can be used as a measure for mitochondrial polarization. Stimulation with TUG-891 resulted in fading of the MTR signal

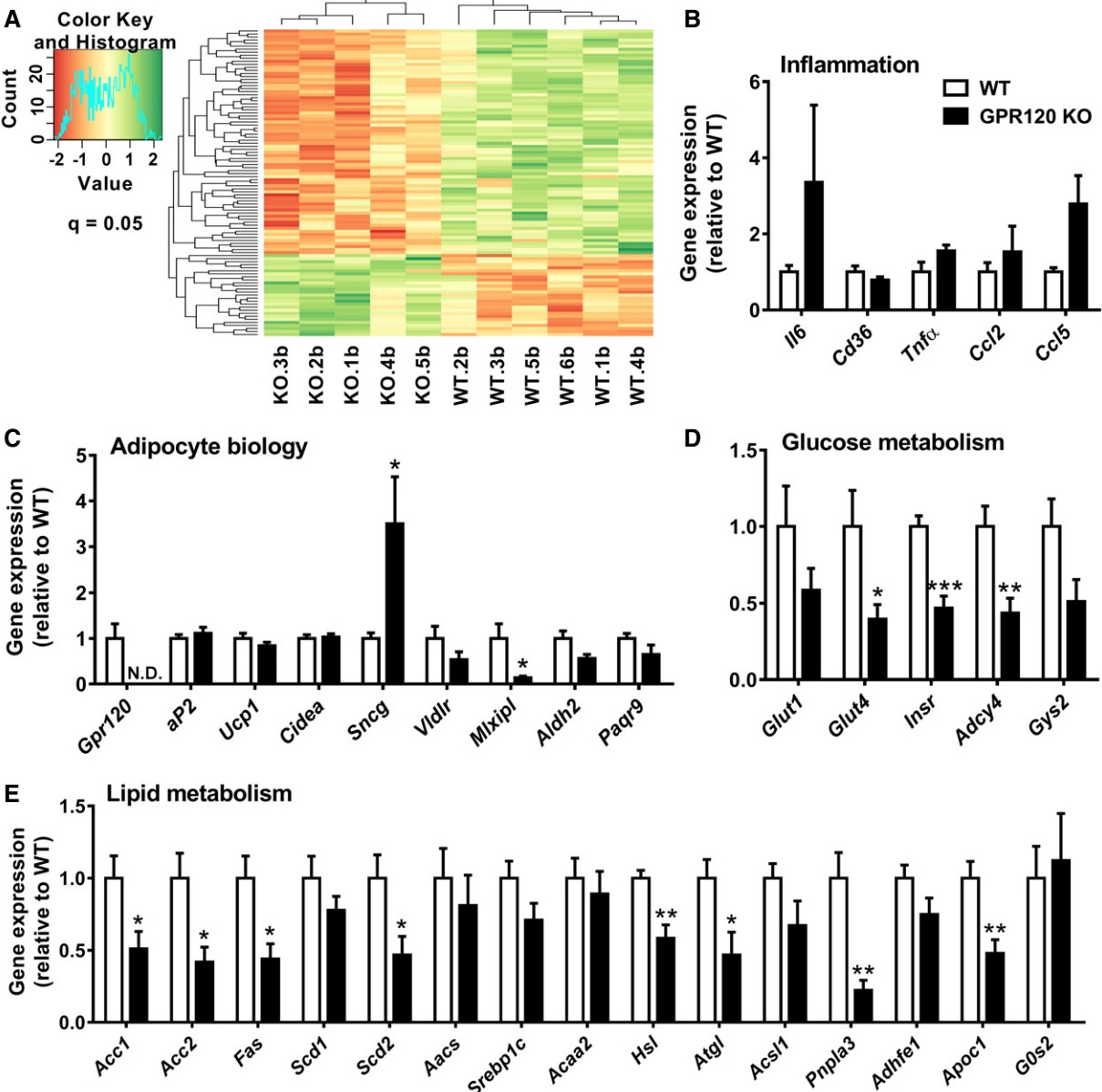

**Figure 5.  GPR120 deficiency alters the expression of genes involved in glucose and lipid metabolism in BAT.**

A    Probe sets for WT and GPR120 KO BAT from microarray analysis are colored according to average expression levels across all samples, with green denoting a higher expression level and red denoting a lower expression level. The probe sets shown in the heat map passed the threshold of absolute value of the logFC > 0.5 and $P$-adjusted < 0.05.

B–E    Expression of genes involved in inflammation, adipocyte biology, glucose metabolism, and lipid metabolism in BAT from GPR120 KO mice ($n$ = 5) and WT littermates ($n$ = 6) was determined through qRT–PCR (N.D. = non-detectable). Data represent means ± SEM. *$P$ < 0.05, **$P$ < 0.01, ***$P$ < 0.001 compared to the WT control group, according to the two-tailed unpaired Student's $t$-test. The exact $P$-value for each significant difference can be found in Appendix Table S5.

while the MTG signal remained intense, indicative of mitochondrial depolarization (Fig 7F). In addition, mitochondria were more fragmented following TUG-891 stimulation (Fig 7G), pointing toward increased mitochondrial fission, which could explain the GPR120-dependent increase in respiration. Of note, the timing of TUG-891-induced changes in mitochondrial morphology coincides with increases in intracellular $Ca^{2+}$, suggesting this effect is mediated through $Ca^{2+}$.

# Discussion

In the current study, we aimed to investigate the therapeutic potential and mechanism of action of GPR120 agonism. We specifically focussed on BAT and demonstrated that the GPR120 agonist TUG-891 increases the activity of brown adipocytes, potentially by stimulating $Ca^{2+}$-induced mitochondrial depolarization and fission. In addition, TUG-891 may act GPR120 independently to activate UCP1

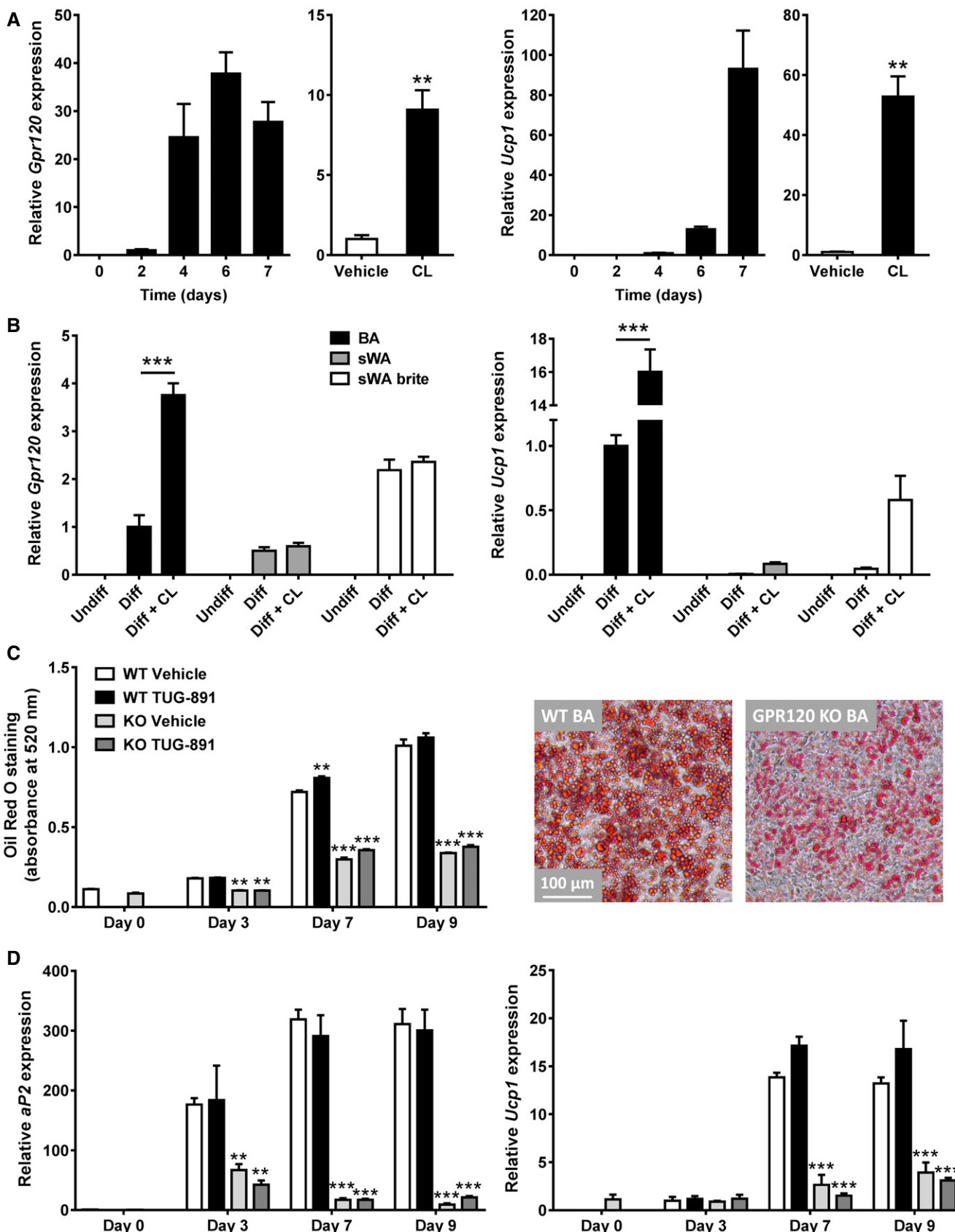

**Figure 6.**

**Figure 6.  GPR120 is involved in differentiation of brown adipocytes and browning of white adipocytes.**

A  Immortalized murine brown adipocytes (*n* = 3) were differentiated for 0, 2, 4, 6, or 7 days after which expression of *Gpr120* and *Ucp1* was determined by qRT–PCR. On day 7, a subset of adipocytes (*n* = 3) was stimulated with CL (10 μM) or vehicle.

B  Expression of *Gpr120* and *Ucp1* was measured in undifferentiated (Undiff), differentiated (Diff), and CL-treated (Diff + CL) brown adipocytes (BA), subcutaneous white adipocytes (sWA), and sWA treated with the browning agent rosiglitazone (sWA brite) (*n* = 3).

C  WT and GPR120 KO brown adipocytes (*n* = 3) were treated with vehicle or TUG-891 (10 μM) throughout differentiation and stained at day 0, 3, 7, and 9 of differentiation with Oil Red O. Absorbance of the staining at 520 nm was quantified. A representative image at day 8 of differentiation was taken with a phase-contrast microscope (Leica) at 20-fold magnification.

D  As in (C), WT and GPR120 KO brown adipocytes (*n* = 3) were treated with vehicle or TUG-891 throughout differentiation to analyze expression patterns of *aP2* and *Ucp1*.

Data information: Data represent means ± SEM. **$P < 0.01$ compared to the vehicle group, ***$P < 0.001$ compared to the WT control group or indicated controls, according to the two-tailed unpaired Student's *t*-test (A) or two-way ANOVA with Dunnett's *post hoc* test (B–D). The exact *P*-value for each significant difference can be found in Appendix Table S5.

and increase uncoupled respiration. These mechanisms could have acted synergistically *in vivo* to induce BAT activation, thereby increasing lipid oxidation and reducing fat mass.

We assessed the therapeutic potential of GPR120 activation by using the agonist TUG-891, a more selective and potent agonist for GPR120 than α-linolenic acid, GW9508, and NCG21 (Shimpukade *et al*, 2012; Hudson *et al*, 2013). Mice treated with TUG-891 exhibited decreased body weight and fat mass, and an increased fat oxidation. These results are in line with a previous study that observed reduced body weight after chronic GPR120 agonist treatment in diet-induced obese mice (Azevedo *et al*, 2016). The increased fat oxidation upon TUG-891 treatment suggested involvement of BAT, as previous studies have demonstrated that selective BAT activation specifically stimulates lipid oxidation (Berbee *et al*, 2015; Schilperoort *et al*, 2016). Indeed, TUG-891 enhanced the uptake of TG-derived FA by BAT, indicating an increased lipid combustion in BAT resulting in a higher need to take up lipids from the circulation. Moreover, lipid droplet content in iBAT and total iBAT weight were decreased in TUG-891-treated mice, also a feature of increased BAT activity. Adipocyte size was decreased in WAT of TUG-891-treated mice, indicative of increased lipolysis in WAT, possibly to release lipids into the circulation to fuel the highly active BAT.

To further elucidate the importance of GPR120 for BAT functionality, we examined expression patterns of *Gpr120 in vivo* and *in vitro*. We found that *Gpr120* is highly expressed in BAT as compared to other tissues. Furthermore, *Gpr120* expression increased during brown adipocyte differentiation and upon treatment with the classical BAT activator CL (Berbee *et al*, 2015). This is in line with previous data showing that cold exposure, the most potent browning stimulus, increases the expression of *Gpr120* in BAT, sWAT, and gWAT (Rosell *et al*, 2014). Rosiglitazone, a PPARγ agonist that stimulates browning (Ohno *et al*, 2012), induced *Gpr120* expression in subcutaneous white adipocytes *in vitro*. A similar effect of browning on *Gpr120* expression was observed earlier upon treatment of 3T3-L1 white adipocytes with the PPAR agonist troglitazone (Gotoh *et al*, 2007). These results demonstrate that like *Ucp1*, *Gpr120* is highly expressed in brown adipocytes and increases during differentiation and browning, signifying an important role for *Gpr120* in BAT physiology. This is further supported by reduced lipid accumulation and *aP2* expression in GPR120-deficient adipocytes, indicative of impaired differentiation. The latter findings concur with previous studies that used siRNA to knockdown *Gpr120* expression in 3T3-L1 cells, which resulted in reduced lipid droplet accumulation and *aP2* expression (Gotoh *et al*, 2007; Liu *et al*, 2012). The expression of genes involved in glucose and lipid metabolism was reduced in BAT from GPR120-deficient mice, suggesting that the uptake and handling of nutrients are less efficient in the absence of GPR120. This is in accordance with the increased uptake of nutrients by BAT upon GPR120 stimulation by TUG-891.

Our findings that expression of *Gpr120* in BAT was highest as compared to other organs and that *Gpr120* expression is induced upon brown adipocyte differentiation were corroborated by

**Figure 7.  TUG-891 increases oxygen consumption by brown adipocytes, mediated by direct UCP1 activation and mitochondrial fragmentation.**

A  Immortalized brown adipocyte (*n* = 5–6) was pretreated with either vehicle or the GPR120 antagonist AH7614 (100 μM) for 30 min, followed by measurement of the basal oxygen consumption rate (OCR) for 15 min in a Seahorse XF24 analyzer. Hereafter, cells were treated with either vehicle or the GPR120 agonist TUG-891 (10 μM) and OCR was measured for another 30 min.

B, C  Representative traces showing the effects of TUG-891 (heavy line) and vehicle (thin line) on oxygen consumption in BAT mitochondria (0.125 mg/ml) from WT (B) and UCP1 KO (C) mice (*n* = 4–5). Additions were mitochondria (M), GDP (1 mM), TUG-891 (successively added in the concentration range of 10–150 μM), and FCCP (1.0–1.4 μM). The breaks in trace indicate periods of chamber re-oxygenation.

D  Immortalized brown adipocytes (*n* = 5–6) were pretreated for 30 min with vehicle or BAPTA-AM (25 μM), after which the OCR was determined in a Seahorse XF24 analyzer. After three baseline measurements, either vehicle or TUG-891 (10 μM) was injected into the wells.

E  WT and GPR120 KO brown adipocytes were incubated with the calcium-sensitive dye Fluo-4-AM for 1 h at RT, followed by live cell imaging with a confocal laser scanning microscope (LSM 510, Zeiss) and stimulation with TUG-891 (10 μM) with or without the presence of AH7614 (100 μM). $F_1/F_0$ represents peak fluorescence divided by baseline fluorescence.

F  Representative images of a brown adipocyte stained with MitoTracker Green FM (125 nM) and MitoTracker Red CMXRos (250 nM) before and after stimulation with TUG-891 (10 μM) for 10 min. Fluorescence intensity of MTR/MTG was determined in TUG-891-treated cells and controls (*n* = 8–9) at baseline and after 10 min of fluorescence imaging, and plotted relative to baseline.

G  Representative images of a brown adipocyte stained with MitoTracker Green FM at baseline (0 min) and 2, 4, and 8 min after TUG-891 stimulation. MitoTracker-stained live cells were imaged using a confocal laser scanning microscope (Leica TCS SP8, Leica Microsystems).

Data information: Data represent means ± SEM. **$P < 0.01$, ***$P < 0.001$ compared to the vehicle group, #$P < 0.05$ compared to the TUG-891 control group (or baseline in Fig 7F), according to the two-tailed unpaired Student's *t*-test. The exact *P*-value for each significant difference can be found in Appendix Table S5.

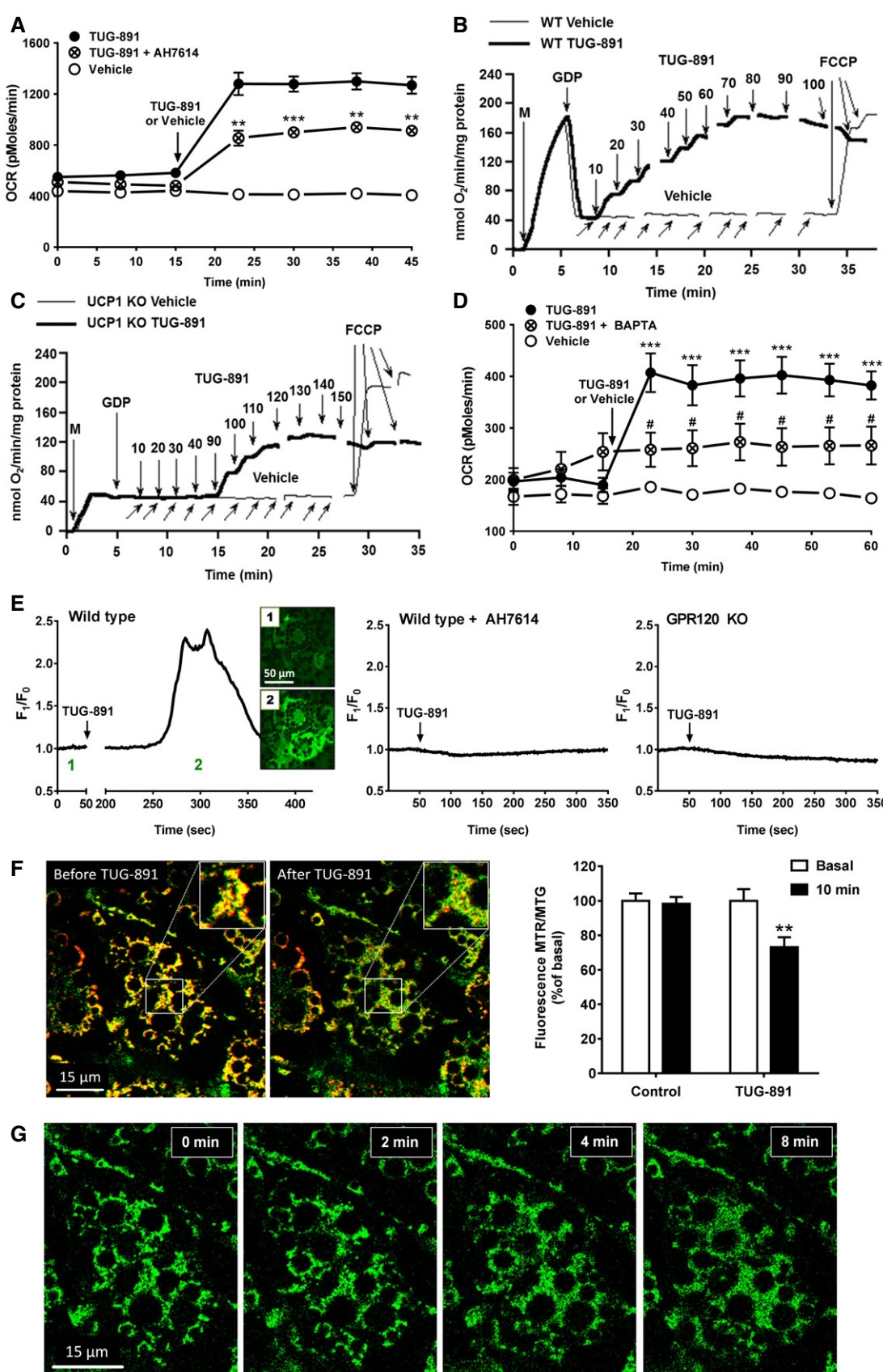

**Figure 7.**

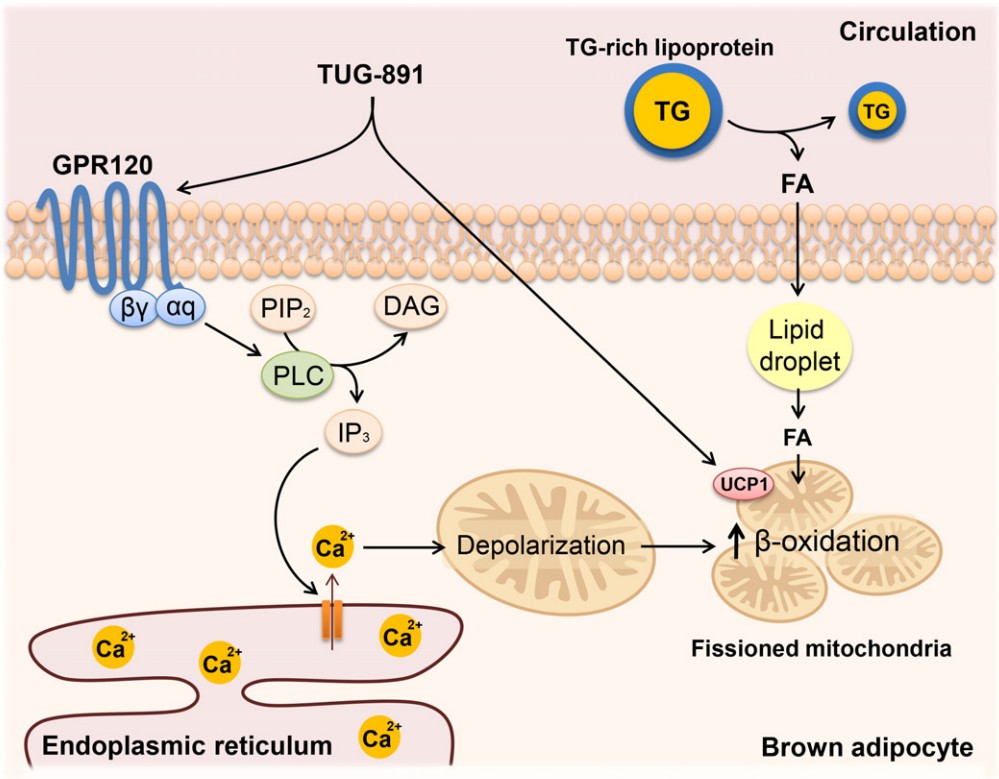

**Figure 8. Proposed mechanism by which the GPR120 agonist TUG-891 activates BAT.**

TUG-891 selectively agonizes the Gαq-coupled GPR120, which activates phospholipase C (PLC). Upon activation, PLC cleaves phospholipid phosphatidylinositol 4,5-bisphosphate (PIP2) into diacyl glycerol (DAG) and inositol trisphosphate (IP3). IP3 triggers the opening of $Ca^{2+}$ channels in the membrane of the endoplasmic reticulum, thereby increasing intracellular $Ca^{2+}$ concentrations. Increased $Ca^{2+}$ leads to depolarization of the mitochondria, and subsequently induction of mitochondrial fission which increases respiration. In addition, TUG-891 directly activates UCP1, further stimulating uncoupled respiration and lipid combustion. As a consequence, the activated brown adipocytes take up fatty acids (FA) from triglyceride (TG)-rich lipoproteins from the circulation, which eventually reduces fat mass.

Quesada-López et al (2016). They showed that the GPR120 agonist GW9508 upregulated thermogenic genes in BAT and increased $O_2$ consumption in mice, without changes in body weight and food intake. However, GW9508 also increased $O_2$ consumption and UCP1 levels in GPR120 KO animals. Potentially, these effects of GW9508 are mediated through GPR40 (Ou et al, 2013), as GW9508 activates both GPR40 and GPR120 and is approximately 100-fold more selective for GPR40 than GPR120 (Briscoe et al, 2006). GPR40 is involved in insulin secretion and glucose metabolism (Itoh et al, 2003; El-Azzouny et al, 2014), and GPR40 KO mice develop obesity, glucose intolerance, and insulin resistance (Kebede et al, 2008). Also, activation of GPR40 has recently been shown to reduce food intake and body weight in mice (Gorski et al, 2017). In our study, food intake was reduced in both wild-type and GPR120 KO mice treated with TUG-891. Therefore, this effect might be mediated through GPR40 instead of GPR120. However, effects of TUG-891 on body weight, fat mass, and fat oxidation were reduced or absent in GPR120 KO mice, confirming that these beneficial metabolic effects of TUG-891 were predominately mediated through GPR120.

We next verified whether the BAT-activating effects of TUG-891 in vivo were a consequence of a direct effect of TUG-891 on brown adipocytes. In line with the acute effect of TUG-891 on the RER and fat oxidation in vivo, TUG-891 acutely increased $O_2$ consumption by brown adipocytes in vitro. This indicates that the effects of TUG-891 in vivo could all have been mediated by direct BAT activation. However, we cannot exclude involvement of other tissues, as GPR120 is not exclusively expressed on brown adipocytes. For example, GPR120 is expressed in the hypothalamus and central agonism of GPR120 has been shown to affect energy metabolism (Auguste et al, 2016; Dragano et al, 2017). However, as carboxylic acids similar to TUG-891 have difficulty penetrating the blood–brain barrier, this is not very likely (Pajouhesh & Lenz, 2005). Also, conflicting reports exist on whether GPR120 plays a role in muscle physiology and metabolism (Oh et al, 2010; Kim et al, 2015). In our study, expression of Gadd45a, Murf1, and Myog in skeletal muscle tissue was mildly affected by TUG-891 treatment. Whether this is an off-target effect or GPR120-mediated effect, which could affect muscle function, remains to be investigated. Future experiments with tissue-specific GPR120 KO mice would be valuable to assess tissue specificity of TUG-891. In addition, it would be interesting to repeat our in vivo experiments at thermoneutrality, to substantiate the link between BAT activation and the observed phenotype.

Using a GPR120 antagonist, we discovered that the TUG-891-induced increase in $O_2$ consumption in brown adipocytes is only

partly mediated by GPR120. The GPR120-independent effect of TUG-891 could be due to direct activation of mitochondrial UCP1. TUG-891 relieves the natural inhibition of UCP1 by GDP (Matthias *et al*, 2000), similar to oleate and other LCFAs (Shabalina *et al*, 2004; Fedorenko *et al*, 2012), thereby leading to increased UCP1 activity and uncoupled mitochondrial respiration. This could explain the moderately decreased fat mass and increased FA uptake by BAT in TUG-891-treated GPR120 KO mice. However, as most metabolic effects of TUG-891 were largely attenuated or abolished in GPR120 KO mice, BAT activation by TUG-891 *in vivo* is mainly dependent on GPR120 signaling.

To investigate through which mechanism GPR120 signaling could increase brown adipocyte activity, several Gαq-coupled signaling pathways were studied: the PI3K/AKT pathway, MAPK/ERK pathway, and signaling through $Ca^{2+}$. Of these, only the intracellular $Ca^{2+}$ availability was proven to be essential for GPR120-mediated activation of brown adipocytes. A recent study showed that $Ca^{2+}$ could increase respiration in brown adipocytes by decreasing the mitochondrial membrane potential (MMP) (Hou *et al*, 2017). Evidently, the β receptor agonist isoprenaline induces $Ca^{2+}$ release from the endoplasmic reticulum of brown adipocytes resulting in mitochondrial depolarization and fission (Hou *et al*, 2017), the latter being a process required for NA-induced uncoupled respiration (Wikstrom *et al*, 2014). Therefore, we studied whether this $Ca^{2+}$-mediated pathway of mitochondrial depolarization and fission could also underlie GPR120-mediated activation of brown adipocytes. Mitochondria were co-stained with the MMP-sensitive MitoTracker CMXRos (MTR) and the MMP-insensitive MitoTracker Green (MTG) (Pendergrass *et al*, 2004), and the relative ratio of MTR/MTG was used as a measure for mitochondrial depolarization (as seen in (Wikstrom *et al*, 2014) in which TMRE was used instead of MTR). TUG-891 stimulation resulted in a reduction in the MTR/MTG ratio, indicative of mitochondrial depolarization. Also, TUG-891 increased mitochondrial fragmentation, presumably secondary to $Ca^{2+}$-induced mitochondrial depolarization. These results suggest that GPR120 signaling could increase metabolic activity of brown adipocytes by stimulation of mitochondrial fission in a $Ca^{2+}$-dependent manner.

We conclude that TUG-891, an agonist of the free FA receptor GPR120, directly stimulates BAT activity via both GPR120-dependent and GPR120-independent mechanisms (Fig 8). As a consequence, lipid uptake and oxidation by BAT increases, eventually reducing body weight and fat mass. Since impaired GPR120 signaling predisposes to obesity in humans (Ichimura *et al*, 2012), obese individuals could benefit from GPR120 activation. Although further studies are needed to investigate the safety of TUG-891 and the potential of GPR120 agonists to activate BAT in humans, we thus anticipate that GPR120 agonism is a promising therapeutic strategy to increase BAT activity, thereby increasing fat oxidation and reducing obesity.

# Materials and Methods

## Animals

To assess effects of TUG-891 on energy metabolism, 8- to 10-week-old male C57Bl/6J mice (Charles River Laboratories) were randomized to receive an intraperitoneal injection with either TUG-891 (35 mg/kg) or 10% dimethyl sulfoxide (DMSO) vehicle dissolved in PBS once daily for 2.5 weeks. TUG-891 was synthesized as previously described (Shimpukade *et al*, 2012) and was of > 99.5% purity, as assessed by HPLC and NMR. Mice were injected 2 h before initiation of the dark phase. To evaluate the specificity of TUG-891 for GPR120, this experiment was also performed in 10- to 14-week-old male GPR120 knockout (KO) mice and wild-type (WT) littermates on a C57Bl/6J background for a total period of 2 weeks. All mice were housed in conventional cages with a 12-h light/dark cycle and had *ad libitum* access to chow diet and water. Mouse experiments were performed in accordance with the Institute for Laboratory Animal Research Guide for the Care and Use of Laboratory Animals after having received approval from the University Ethical Review Board (Leiden University Medical Center, Leiden, The Netherlands).

To evaluate *Gpr120* gene expression in various tissues, 12-week-old FVB/N female mice were sacrificed by cervical dislocation and organs were collected. Mice were housed in conventional cages with a 12-h light/dark cycle and had *ad libitum* access to chow diet and water, and experiments were carried out in accordance with UK Home Office regulations.

For experiments in which mitochondrial respiration was measured, mitochondria were isolated from 8- to 10-week-old male UCP1 KO mice (progeny of those described in Enerback *et al*, 1997) backcrossed to C57Bl/6J mice and wild-type C57Bl/6J mice. Mice were housed in conventional cages with a 12-h light/dark cycle and had *ad libitum* access to chow diet and water, and experiments were carried out in accordance with the Animal Ethics Committee of the North Stockholm region in Sweden.

## Food intake, body weight, and body composition measurements

At the indicated time points, food intake and body weight of mice were measured with a scale, and lean and fat mass with an EchoMRI-100-analyzer.

## Indirect calorimetry

Indirect calorimetry was performed in fully automated metabolic cages (LabMaster System, TSE Systems) during the first week of treatment. After 3 days of acclimatization, $O_2$ consumption ($VO_2$), $CO_2$ production ($VCO_2$), and caloric intake were measured for 5 consecutive days. Total EE was estimated from the $VO_2$ and resting energy requirement. Carbohydrate oxidation was calculated using the formula $((4.585*VCO_2) - (3.226*VO_2))*4$, in which the 4 represents the conversion from mass per time unit to kcal per time unit (Peronnet & Massicotte, 1991). Similarly, fat oxidation was calculated using the formula $((1.695*VO_2) - (1.701*VCO_2))*9$. Physical activity was measured with infrared sensor frames.

## Tissue histology and immunohistochemistry

Formalin-fixed interscapular BAT (iBAT), subcutaneous WAT (sWAT), and gonadal WAT (gWAT) were dehydrated in 70% EtOH, embedded in paraffin, and cut into 5-μm sections. Sections were stained with hematoxylin and eosin (H&E) using standard protocols. UCP1 staining was performed as previously described (Berbee *et al*, 2015). In short,

sections were treated with 3% $H_2O_2$ for 30 min and boiled in citrate buffer (10 mM, pH 6) for 10 min. Slides were blocked with 1.3% normal goat serum, incubated overnight at 4°C with rabbit monoclonal anti-UCP1 antibody (1:400, Abcam) followed by 1-h incubation with biotinylated goat α-rabbit secondary antibody (Vector Labs). Immunostaining was amplified using Vector Laboratories Elite ABC kit (Vector Labs) and visualized with Nova Red (Vector Labs). Counterstaining was performed with hematoxylin. All sections were digitalized with Philips Digital Pathology Solutions (PHILIPS Electronics) for morphological measurement. White adipocyte size, iBAT lipid droplet content, and UCP1 expression (relative UCP1 staining per area) were quantified using ImageJ software (Version 1.50).

## Plasma triglycerides

After 6 h of food withdrawal, blood was collected from the tail vein in paraoxon-coated capillaries, and plasma levels of TG were determined using an enzymatic kit (Roche Diagnostics)

## RNA isolation, cDNA synthesis, and qRT–PCR

Tissues or cells were dissolved in TRIzol RNA isolation reagent (Thermo Fisher) following the manufacturer's protocol. The RNA concentration was determined with a NanoDrop spectrophotometer (Thermo Fisher). For removal of genomic DNA, samples were treated with DNase I (Sigma), after which total RNA was reverse-transcribed with M-MLV Reverse Transcriptase (Sigma). The qRT–PCR was performed with a SYBR Green kit (Sigma) on a 7500 Fast RT–PCR System (Applied Biosystems). Primer sequences are listed in Appendix Table S1. mRNA expression of genes of interest was normalized to mRNA expression of the housekeeping genes *L19*, *β-Actin*, and/or *β2-microglobulin*.

## Protein isolation and Western blot analysis

Cells were stimulated with TUG-891 (10 μM) for 5 min to assess phosphorylation of ERK and AKT after which they were lysed in ice-cold RIPA buffer (50 mM Tris–HCL pH 8, 1 mM EDTA, 150 mM NaCl, 1% NP-40, 0.5% sodium deoxycholate, 0.1% SDS) containing protease and phosphatase inhibitor cocktails (Roche). Homogenates were centrifuged, and protein content of the supernatant was determined using a Coomassie Protein Assay Kit (Thermo Fisher). After heating the samples (5 min, 95°C), 20 μg of protein was separated by 12% SDS–PAGE, followed by transfer to a PVDF membrane using the Trans-Blot Turbo Transfer System (Bio-Rad). Membranes were blocked with 5% milk, incubated overnight at 4°C with protein-specific primary antibody followed by incubation for 1 h with horseradish peroxidase (HRP)-conjugated secondary antibodies (Goat anti-Rabbit HRP, Dako P0448 at 1:2,000 or Promega W4018 at 1:1,000). Primary antibodies used were rabbit anti-pHSL563 (Cell Signaling #4139 at 1:1,000), rabbit anti-UCP1 (Sigma U6382 at 1:4,000), rabbit anti-tubulin (Cell Signaling #2148 at 1:1,000), rabbit anti-pPKA substrate (Cell Signalling #9612 at 1:1,000), rabbit anti-pERK 1/2 (Cell Signaling #9101 at 1:500), rabbit anti-ERK 1/2 (Cell Signaling #4695 at 1:5,000), rabbit anti-pAKT (Ser 473) (Cell Signaling #9271 at 1:1,000), rabbit anti-AKT (Cell Signaling #9272 at 1:1,000), and mouse anti-β-Actin HRP (Santa Cruz sc-47778 at 1:5,000). Bands were visualized using Amersham ECL Prime Western Blotting Detection Reagent (GE Healthcare) and quantified using ImageJ software (Version 1.50).

## Clearance of radiolabeled lipoprotein-like emulsion particles and glucose

Glycerol tri[³H]oleate ([³H]TO)-labeled lipoprotein-like TG-rich emulsion particles (80 nm) were prepared and characterized as described previously (Rensen *et al*, 1995), and [¹⁴C]deoxyglucose ([¹⁴C]DG) was added (ratio ³H:¹⁴C = 4:1). Mice were fasted for 6 h and injected with 200 μl of emulsion particles (1 mg TG per mouse) via the tail vein, 1 h after onset of the dark phase (i.e., 3 h after injection of TUG891 or vehicle). After 15 min, mice were sacrificed by cervical dislocation and perfused with ice-cold PBS through the heart. Thereafter, organs were harvested and weighed, and dissolved overnight at 56°C in Tissue Solubilizer (Amersham Biosciences). The uptake of [³H]TO- and [¹⁴C]DG-derived radioactivity was quantified and expressed per gram of wet tissue weight or per organ for organs that could be taken out quantitatively.

## Microarray experiments

iBAT from GPR120 KO mice (Godinot *et al*, 2013) and WT littermates with similar body weights were analyzed. Global mRNA expression was measured using Illumina bead chip. Whole-genome expression was profiled using MouseWG-6 v2.0 Expression Bead-Chips (Kuhn *et al*, 2004). The summary-level data were processed using the R packages lumi 2.10.0, lumiMouseAll.db 1.18.0, and lumiMouseIDMapping 1.10.0 using nuID annotations (Du *et al*, 2007). The data were normalized using quantile normalization. Differentially expressed genes (DEGs) between WT and KO samples were detected based on a moderated *t*-test using limma on the normalized data, removing unexpressed genes. The normalized expression data were filtered using the absolute value of logFC < 0.5 and adjusted *P*-value < 0.05, converted into z-scores, and heatmap.2 was used to visualize the data. DEGs with *P* < 0.05 were submitted to DAVID (Database for Annotation, Visualization and Integrated Discovery) for functional classification by using RefSeq mRNA accession numbers. Functional clusters were considered significant for FDR (false discovery rate) < 0.01 (da Huang *et al*, 2009).

## Cell culture

Previously described immortalized cell lines of primary cultures of BAT and sWAT were used for experiments (Rosell *et al*, 2014). Immortalized GPR120 KO brown adipocytes were generated in the same way as these primary culture cell lines, *that is,* by retroviral-mediated transformation of SV40 large T-antigen. Brown and white preadipocytes were cultured in DMEM/F12 (Sigma) supplemented with 10% fetal bovine serum (FBS) and penicillin–streptomycin (Sigma). Preadipocytes were differentiated with induction medium for 2 days and with maintenance medium for 6 days. The constituents of this medium and their concentrations are listed in Appendix Table S2. To assess effects of browning of white adipocytes on gene expression, the induction medium was modulated to contain 5 μM rosiglitazone, after which the cells received 1 μM of rosiglitazone during the first 4 days of maintenance. Adipocytes were used for experiments on day 7–9 of differentiation.

## Oil Red O staining

Differentiated immortalized brown adipocytes were fixed with 4% paraformaldehyde (15 min, RT) and rinsed with 60% isopropanol. Cells were stained with 0.15% Oil Red O (Sigma) in 60% isopropanol (30 min, RT), after which they were washed with 60% isopropanol. Images were taken with a phase-contrast microscope (Leica).

## Cellular oxygen consumption measurements

Seahorse Bioscience XF24 extracellular flux analyzer (Seahorse Bioscience) was used to measure the OCR in differentiated brown adipocytes. On day 7 of differentiation, cells were trypsinized and seeded in a 24-well Seahorse Bioscience assay plate. The next day, cells were pretreated with BAPTA-AM (25 μM; Thermo Fisher), U0126 (10 μM; Promega), or AKT 1/2 kinase inhibitor (10 μM; A6730, Sigma) for 30 min (37°C, without $CO_2$) before starting Seahorse analysis. Vehicle (DMSO), TUG-891, and/or CL316243 (CL) were preloaded in the reagent delivery chambers and pneumatically injected into the wells after three baseline measurements (to a final concentration of 10 μM TUG-891 or CL). Cellular $O_2$ consumption was measured in real time every 7 min.

## Isolation of mitochondria

Brown fat mitochondria were prepared as previously described (Cannon & Nedergaard, 2008; Shabalina *et al*, 2010). Routinely, on each experimental day, three mice were anaesthetized for 1–2 min by a mixture of 79% $CO_2$ and 21% $O_2$, and decapitated. The interscapular, axillary, and cervical BAT depots were dissected out, cleaned from WAT, and pooled in ice-cold isolation buffer, SHE consisting of 250 mM sucrose, 10 mM HEPES (pH 7.2), 0.1 mM EGTA, and 2% (w/v) FA-free BSA (10775835001 Roche Diagnostics GmbH). Throughout the isolation process, the tissue was kept at 0–4°C. Tissue was minced with scissors, homogenized in SHE buffer with a motorized Potter-Elvehjem Teflon pestle, filtered through cotton gauze, and centrifuged at 8,800 *g* for 10 min. The supernatant with the floating fat layer was discarded. The resuspended homogenate was centrifuged at 800 *g* for 10 min, and the resulting supernatant was centrifuged at 8,800 *g* for 10 min. The resulting mitochondrial pellet was resuspended in 100 mM KCl, 20 mM $K^+$-Tes (pH 7.2), and centrifuged again at 8,800 *g* for 10 min. The final mitochondrial pellets were resuspended by hand homogenization in a small glass homogenizer in the same medium to yield a concentration of roughly 25–35 mg/ml mitochondrial protein. The concentration of mitochondrial protein was measured using fluorescamine (Fluram, 47614 Sigma-Aldrich; Udenfriend *et al*, 1972) with BSA as a standard. Mitochondria were stored on ice, and aliquots were removed as required during functional analyses.

## Mitochondrial oxygen consumption measurements

For oxygen consumption measurements, isolated brown fat mitochondria (0.25 mg protein) were added to 2.0 ml of a continuously stirred incubation medium consisting of 100 mM KCl, 20 mM $K^+$-Tes (pH 7.2), 2 mM $MgCl_2$, 1 mM EDTA, 4 mM KPi, 3 mM malate, 5 mM pyruvate (Sigma-Aldrich), and 0.1% FA-free BSA. Oxygen consumption rates were monitored using an O2k-MultiSensor System (Oroboros Instruments) in a sealed incubation chamber at 37°C. During prolonged recording, re-oxygenation of respiratory buffer was performed by unsealing of chamber. Basal respiration was measured in the presence of 1–3 mM GDP (dissolved in 20 mM Tes (final pH 7.2), G7127 Sigma-Aldrich). Maximal oxygen consumption rates (respiratory capacity) were obtained by addition of the ionophoric uncoupling agent FCCP (C2920, Sigma-Aldrich) at a final concentration of 1.0–1.4 μM. TUG-891 was dissolved in DMSO at a stock concentration of 100 mM or 20 mM and used for titration by adding 1–2 μl to 2-ml chamber.

## Calcium mobilization assays

Differentiated cells were incubated for 1 h at RT with the calcium-sensitive dye Fluo-4-AM (Invitrogen) in Krebs–Ringer bicarbonate buffer (Sigma). Hereafter, the cells were washed twice with buffer, followed by live cell imaging with a confocal laser scanning microscope (LSM 510, Zeiss) and stimulation with TUG-891 (10 μM), with or without preincubation with the GPR120 antagonist AH7614 for 5 min (100 μM; Tocris) or the Gαq inhibitor YM-254890 for 30 min (0.1 μM, Alpha Laboratories).

## MitoTracker experiments

Differentiated adipocytes were incubated for 30 min with Mito-Tracker Green FM (125 nM; Thermo Fisher) and MitoTracker Red CMXRos (250 nM; Thermo Fisher) in DMEM/F12 (Sigma) without FBS. Hereafter, the medium was changed and live cells were imaged using a confocal LSM (Leica TCS SP8, Leica Microsystems). Adipocytes were stimulated with TUG-891 (10 μM), followed by live cell imaging for 10 min to monitor mitochondrial morphology. Control cells were monitored to correct for potential photobleaching, and corrected total cell fluorescence (CTSF, integrated density − (area of selected cell × mean fluorescence of background)) of 8–9 cells per condition was quantified using ImageJ.

## cAMP measurements

Measurement of whole cell cAMP was carried out with the cAMP dynamic 2 kit (Cisbio Bioassays, 62AM4PEC) as per manufacturer's instructions. Cells were pretreated with phosphodiesterase inhibitor 3-isobutyl-1-methylxanthine (IBMX, 0.5 mM, 5 min) prior to ligand treatment and lysed in 0.1 M HCl/0.1% Triton X-100. All cAMP concentrations were corrected for protein levels.

## Statistical analysis

All data are expressed as mean ± SEM. Statistical analysis was performed using two-tailed unpaired Student's *t*-test or ANOVA with Tukey's *post hoc* test using SPSS Statistics (Version 23.0). Differences between groups were considered statistically significant at $P < 0.05$.

## Data availability

The microarray data from this publication have been deposited to NCBI's Gene Expression Omnibus (Edgar *et al*, 2002) and assigned

## The paper explained

### Problem

Activation of brown adipose tissue (BAT) could be a promising strategy to promote energy expenditure and combat obesity and related disorders. A potential target to activate BAT is G protein-coupled receptor 120 (GPR120), which is highly expressed in BAT and associated with obesity in humans. However, the therapeutic potential of GPR120 agonism and GPR120-mediated signaling in BAT remain to be elucidated.

### Results

Here, we show that activation of GPR120 by the selective agonist TUG-891 acutely increased fat oxidation and reduced body weight and fat mass in mice. These effects coincided with decreased brown adipocyte lipid content and increased nutrient uptake by BAT, demonstrating that increased BAT activity could underly the improved metabolism of these mice. Mechanistically, TUG-891 activated brown adipocytes *in vitro* through GPR120-dependent and GPR120-independent mechanisms. TUG-891 stimulated the Gαq-coupled GPR120, resulting in intracellular calcium release, mitochondrial depolarization, and mitochondrial fission. In addition, TUG-891 activated mitochondrial UCP1. These mechanisms could have acted synergistically *in vivo* to stimulate mitochondrial respiration in BAT and increase energy expenditure.

### Impact

Since impaired GPR120 signaling predisposes to obesity in humans, obese individuals could benefit from GPR120 activation. Indeed, our data suggest that GPR120 agonism is a promising strategy to increase lipid combustion and reduces obesity.

the GEO Series accession number GSE97145 (http://www.ncbi.nlm.nih.gov/geo/query/acc.cgi?acc = GSE97145).

**Expanded View** for this article is available online.

## Acknowledgements

This work was supported by the Biotechnology and Biological Sciences Research Council (BB/H020233/1 & BB/P008879/1), the EU FP7 project DIABAT (HEALTH-F2-2011-278373), the Genesis Research Trust, the Danish Council for Strategic Research (11-116196), and by personal grants from the Board of Directors of Leiden University Medical Center, the Dutch Heart Foundation, and the Leiden University Fund to M.S. In addition, this work was supported by Eli Lilly and Company through the Lilly Research Award Program, the Netherlands Cardiovascular Research Initiative: an initiative with support of the Dutch Heart Foundation (CVON2011-9 GENIUS to P.C.N.R.), and the Rembrandt Institute of Cardiovascular Science (RICS to P.C.N.R.). P.C.N.R. is an Established Investigator of the Dutch Heart Foundation (2009T038). We thank Karsten Kristiansen and Tao Ma (Laboratory of Genomics and Molecular Biomedicine, Dept. of Biology, Faculty of Science, University of Copenhagen, Denmark) for assistance in generation of the GPR120 KO brown adipocyte cell line, and Claire A Mitchell (Computing and Advanced Microscopy Development Unit, Warwick Medical School, University of Warwick, Coventry, UK) for assistance with confocal imaging. We thank Barbara Cannon and Jan Nedergaard (Dept. of Molecular Biosciences, The Wenner-Gren Institute, Stockholm University, Stockholm, Sweden) for their input regarding data on mitochondrial respiration and UCP1 activation. We thank Lianne van der Wee-Pals, Trea Streefland, and Chris van der Bent (Div. of Endocrinology, Dept. of Medicine, LUMC, Leiden, The Netherlands) for their excellent technical assistance.

## Author contributions

MS performed experiments, analyzed data, and drafted the manuscript. ADvD, GH, IGS, AO, ACH, LHD, IMM, NC, and SK performed experiments, and Y-WC performed bioinformatic analysis. SD provided GPR120 KO BAT used for microarray analysis. ARM and TC provided GPR120 KO mice used for experiments with TUG-891. BS synthesized TUG-891. TU provided tool compounds and contributed to the design of the study. ADvD, PCNR, and MC helped to conceptualize the project and supervised the project. All authors critically reviewed the manuscript.

## Conflict of interest

The authors declare that they have no conflict of interest.

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
