## [Review Process File · EMBO Molecular Medicine]

The GPR120 agonist TUG-891 promotes metabolic health by stimulating mitochondrial respiration in brown fat

Maaike Schilperoort, Andrea D. van Dam, Geerte Hoeke, Irina G. Shabalina, Anthony Okolo, Aylin C. Hanyaloglu, Lea H. Dib, Isabel M. Mol, Natarin Caengprasath, Yi-Wah Chan, Sami Damak, Anne Reifel Miller, Tamer Coskun, Bharat Shimpukade, Trond Ulven, Sander Kooijman, Patrick C.N. Rensen, Mark Christian

Review timeline:

Submission date:	23 May 2017
Editorial Decision:	28 June 2017
Revision received:	20 October 2017
Editorial Decision:	16 November 2017
Revision received:	20 December 2017
Accepted:	21 December 2017

Editor:

Transaction Report:

1st Editorial Decision

28 June 2017

Thank you for the submission of your manuscript to EMBO Molecular Medicine. We have now heard back from the three referees whom we asked to evaluate your manuscript.

You will see from the set of comments pasted below that all three referees find the study of interest. However, while referee 2 is more positive, referees 1 and 3 make pertinent observations that must be addressed experimentally as much as possible to strengthen the conclusions, i.e. GPR120 signalling in brown adipocytes should be looked into, as well as adipogenesis and lipolysis as suggested. Additional discussion and phrasing are also commented upon.

We would welcome the submission of a revised version within three months for further consideration and would like to encourage you to address all the criticisms raised as suggested to improve conclusiveness and clarity.

***** Reviewer's comments *****

Referee #1 (Remarks):

Schilperoort et al. investigate the role of GPR120 in BAT. They find that TUG-891, a selective agonist for GPR120, reduces body weight in mice and promotes adipogenesis in brown adipocytes. Interestingly, they find that O₂ consumption in brown adipocytes is induced in a calcium-dependent manner. This is a very interesting ms. However, several crucial points have to be addressed - especially in the light of previously published papers on GPR120 and Gq signaling in BAT.

- The authors observe lower EE, but increased fat oxidation in the TUG treated mice. However, this was calculated only by a formula. Thus, it would be very informative to analyse lipogenesis and lipolysis markers (since the authors postulate that lipogenesis is increased). Thermogenic markers in BAT and WAT (GPR120 implicated in browning) should be measured, as well as serum TG levels.

- GPR120 signaling in brown adipocytes is still not clear. The authors should use the in vitro model to address this crucial point:

1. GPR120 KO cells differentiate less. It is unclear why. If GPR120 is Gq-coupled, one would expect the opposite. How many wt and ko mice were used to isolate cells, how many independent biological replicates were performed for the differentiation experiments? To address this important point, the authors should at least study the effect of chronic TUG treatment during brown adipocyte differentiation in wt and ko cells (adipogenic and thermogenic markers as well as Oil Red O have to be quantified).
2. Does GPR120 signal via Gs in brown adipocytes? Acute treatment with TUG and its effect on cellular cAMP should be measured.

Minor points:

- Make sure that the recent literature on Gq signaling and GPCRs in BAT is cited correctly.
- Hudson et al., 2013, Mol Pharmacol show that TUG-891 is a potent agonist of FFA4 with only limited selectivity for mouse FFA1, complicating its use in vivo. Therefore the authors use GPR120 knockout mice. They state (line252) "In GPR120 KO mice, TUG-891 failed to significantly reduce total body weight (Fig. 3A) and fat mass (Fig 3B),..." The authors should rephrase this, because TUG clearly reduces both body weight and fat mass, albeit not significantly.
- TUG treated mice have reduced food intake. However, the authors did not discuss whether increased fat turnover could be centrally induced as well. This should be discussed. A tissue-specific deletion of GPR120 only in BAT would be nice to have, but is clearly beyond the scope of this study.

Referee #2 (Remarks):

The paper describes targeting of Gpr120 to increase BAT function and thereby improve metabolism. Use of a Gpr120 agonist leads to a protection from diet induced obesity together with an improved metabolism. They furthermore show that activation of Gpr120 leads to increased brown fat adipogenesis in a Ca dependent manner.

The paper is technically well done and presented. It has to be noted that the effect of Gpr120 has already been described by the group of Villaroya, the novelty and clinical translation of this work is based on the use of a Gpr120 agonist which might also be utilized clinically. Therefore, in my opinion it fits very well to the scope of EMBO Molecular Medicine.

There are a few points which need to be addressed in my opinion:

1. The data in Fig.1 needs to be normalized to lean body mass. This might actually change the statement about reduced utilization of glucose.
2. The data presented in Fig. 2 on inguinal WAT suggests the presence of more beige cells, is this correct. Did the authors look at gene expression of thermogenic genes? How do the authors envisage that these cells are formed (see point 5) and do the authors expect that this contributes to the overall phenotype. This should be discussed in detail (see also point 4).
3. The food intake is interesting especially since the use of ko mice shows a similar trend in response to treatment (albeit not significant), suggesting that at least a small part of the phenotype is mediated by off-target effects. This should be discussed.
4. In general, it is problematic to make the link between BAT activation and the observed phenotype without experiments at thermoneutrality. I am happy that the authors have phrased their manuscript very carefully in that regard, nevertheless this point should be mentioned.
5. The role of Gpr120 in adipogenesis is problematic in my opinion. The only data which supports this notion is the Ap2 expression and maybe the fact that there are more brown adipocytes. Based on the presented data the authors should be able to calculate the overall number of brown adipocytes in BAT from their average cell size data and organ weight. Is this number increased? If yes it would strengthen their point on adipogenesis, if not this part should either be removed or discussed very critically.
6. From my point of view, it looks like TUG might mainly regulate BAT activity (based on the outcome of the calculation in point 6). This should be better reflected in the abstract.

Referee #3 (Remarks):

The manuscript by Schilperoort and colleagues presents results of investigations of the effects of the GPR120 agonist TUG891 on mouse metabolism and particularly on brown adipose tissue. Significantly similar data were recently reported from Villarroya's laboratory, which agree with many of the observations here. There are two more unique points, one of which is a concern, the other potentially of interest provided the necessary controls confirm an effect.

1) The first point that worries me is the decrease in lean body mass that the authors report but subsequently ignore, since they say that the loss of lean is only 10 %, while that of lipid is much greater. While this is mathematically correct, I think the authors should be considerably more concerned, since most of the lean mass is unlikely to change (bones, brain, kidney etc.). They demonstrate a decrease in liver weight but have apparently not measured skeletal muscle. This would be very important to measure to evaluate potential negative effects of the treatment.

2) The second point refers to the observation of an acute stimulation of respiration by TUG in isolated adipocytes. This could be an interesting observation but its validity must be checked. There are several possible reasons for this result. The agonist could stimulate increases in cyclic AMP and the respiration would then follow as it does with e.g. norepinephrine. This is probably not very likely. However, was the effect inhibited by AH7614? If so, this would add credibility to it being a somewhat specific effect.

The agonist could be a direct activator of UCP1, in a similar manner to e.g. fatty acids and retinoic acid. This would be interesting and potentially of therapeutic significance. However, to investigate this further, the authors need to convince me that the effect is not an artifact of the system. Thus, they should repeat the experiments under conditions as indicated in e.g. Li et al. EMBO Rep. 2014 in the presence of 2 % BSA. It would also help if UCP1 KO cells could be used to demonstrate some specificity. Essentially all carboxylic acids can uncouple any mitochondria given sufficient concentration!

Other points:

3) The observation of an increase in intracellular calcium following TUG treatment in isolated adipocytes is an entirely expected response to a Gq-coupled agonist, similarly to the response to an alpha1-adrenergic agonist. Note however that negative effects of BAPTA do not show that the increased oxygen consumption is stimulated through calcium, merely that cytosolic and mitochondrial calcium homeostasis is necessary for most responses in cells and that matrix calcium is required for several citric acid cycle dehydrogenases.

4) The authors note on p. 14 that "little is known about the effects of Ca in brown adipocytes"; this is hardly a scientifically valid statement. If the authors put in "brown fat" and "calcium" in PubMed, they will obtain \approx 250 references, many of which, starting in the earlier 1970s through to the late 1990s, investigate a role of calcium in brown fat mitochondria and isolated adipocytes!

5) There is also something of a conundrum regarding energy expenditure: the wildtype mice lose body weight and decrease food intake without a change in energy expenditure, while the KO mice also decrease food intake but do not decrease body weight. Any comment?

6) Regarding Fig. 4, the authors should show the uptake per organ/depot since the decrease in weight is presumably only triglyceride so that the active components are still present, such that even an unchanged actual uptake per depot would appear to be an increase given per g tissue.

7) The reference to Quesada-López is incomplete.

1st Revision - authors' response

20 October 2017

(begins on next page)

Detailed response to the reviewers' comments

Referee #1:

Schilperoort et al. investigate the role of GPR120 in BAT. They find that TUG-891, a selective agonist for GPR120, reduces body weight in mice and promotes adipogenesis in brown adipocytes. Interestingly, they find that O₂ consumption in brown adipocytes is induced in a calcium-dependent manner. This is a very interesting ms. However, several crucial points have to be addressed - especially in the light of previously published papers on GPR120 and Gq signaling in BAT.

- The authors observe lower EE, but increased fat oxidation in the TUG treated mice. However, this was calculated only by a formula. Thus, it would be very informative to analyse lipogenesis and lipolysis markers (since the authors postulate that lipogenesis is increased). Thermogenic markers in BAT and WAT (GPR120 implicated in browning) should be measured, as well as serum TG levels.

Authors' reply:

We have now measured plasma TG levels in the mice, and observed increased TG levels in TUG-891-treated mice (Appendix Fig S3A). Potentially, this is due to increased lipolysis in WAT to fuel to highly active BAT in these animals.

In addition, we measured expression of the lipolytic genes *Atgl* and *Hsl*, lipogenic genes *Fasn*, *Acc1*, *Acc2*, *Dgat2* and *Scd1*, proliferation genes *Ccna*, *Ccnb* and *Mki67* and the thermogenic genes *Ucp1*, *Cidea*, *Ppara* and *Pparg*, in BAT and/or WAT (Appendix Fig S3F-H). Expression of most genes was unaffected by TUG-891 treatment in BAT.

In WAT, lipogenesis markers were downregulated while proliferation markers were upregulated. Additionally, *Ucp1* expression was markedly increased in gWAT (Appendix Fig S3H), suggesting browning of this fat depot. UCP1 staining of adipose tissues confirmed increased UCP1 levels in gWAT (Appendix Fig S4).

Changes made to the manuscript:

- These data have been added to the Results section (page 11, lines 324-329):

“Plasma TG levels were increased at endpoint, possibly as a result of increased lipolysis (Appendix Fig S3A). Protein (Appendix Fig S3B-E) and gene (Appendix Fig S3F) expression of markers for lipolysis, adipogenesis, proliferation and thermogenesis were largely unaffected in BAT. However, Ucp1 gene expression (Appendix Fig S3H) and protein staining (Appendix Fig S4) was increased in gWAT of TUG-891-treated animals, suggesting GPR120-mediated browning.”

- GPR120 signaling in brown adipocytes is still not clear. The authors should use the in vitro model to address this crucial point:

1. GPR120 KO cells differentiate less. It is unclear why. If GPR120 is Gq-coupled, one would expect the opposite. How many wt and ko mice were used to isolate cells, how many independent biological replicates were performed for the differentiation experiments? To address this important point, the authors should at least study the effect of chronic TUG treatment during brown adipocyte differentiation in wt and ko cells (adipogenic and thermogenic markers as well as Oil Red O have to be quantified).

Authors' reply:

Indeed, a recent study showed that the Gq signaling pathway can have a negative effect on brown adipogenesis (Klepac *et al*, 2016). However, the fact that multiple GPCRs can be coupled to the same type of G-protein does not necessarily mean they have the same mechanism of action. There have been various examples in literature in which Gq coupled receptors have been found to have a positive effect on thermogenic activity of brown adipocytes (Ootsuka & Blessing, 2006, Zhao *et al*, 1997). Also, two independent studies have already shown an impaired differentiation of 3T3-L1 cells after knockdown of *Gpr120* (Gotoh *et al*, 2007, Liu *et al*, 2012). Their results were very comparable with ours, also showing reduced lipid droplet accumulation and *aP2* expression in GPR120-deficient adipocytes. We performed our experiments with three biological replicates per condition. According to the reviewer's suggestion, we now also studied effects of chronic TUG-891 treatment in wildtype and GPR120 KO cells (Fig 6C&D). Again, we observed a decreased *aP2* expression in GPR120 KO adipocytes as compared to wildtype cells. In addition, *Ucp1* expression and lipid droplet accumulation (quantified Oil Red O staining) were decreased in KO cells throughout the differentiation period. Continuous treatment with TUG-891 tended to increase *Ucp1* expression after 7 or 9 days of differentiation in wildtype cells, but not in KO cells.

Changes made to the manuscript:

- These data have been added to the Results section (page 13, lines 398-405):

"To study whether GPR120 is directly involved in adipocyte differentiation, brown adipocyte cell lines were generated from WT and GPR120 KO mice. Both cell lines differentiated to mature brown adipocytes when exposed to a standard hormone differentiation treatment. However, GPR120 KO adipocytes accumulated a lower amount of lipids as evidenced by Oil Red O staining (Fig 6C), and exhibited lower expression of the adipocyte differentiation marker aP2 and Ucp1 (Fig 6D), suggesting impaired differentiation in GPR120 KO cells. Treatment with TUG-891 throughout differentiation tended to increase Ucp1 expression in WT but not GPR120 KO cells (Fig 6D)."

2. Does GPR120 signal via Gs in brown adipocytes? Acute treatment with TUG and its effect on cellular cAMP should be measured.

Authors' reply:

To our knowledge, there is no evidence that GPR120 can signal via Gs in any cell type. To confirm whether this is true for brown adipocytes, we have treated cells with vehicle, TUG-891, or forskolin and measured intracellular cAMP levels. As expected, we found that our positive control forskolin increased cAMP levels, while TUG-891 did not (Appendix Fig S9D).

Also, phosphorylation of PKA substrates and HSL (Ser563) was decreased in BAT of TUG-891-treated mice compared to controls (Appendix Fig S3C-E), suggesting cAMP signaling does not contribute to effects of TUG-891 *in vivo*.

Changes made to the manuscript:

- *In vitro* data concerning cAMP has been added to the Results section (page 14, lines 430-432):

"To ensure that GPR120 is Gαq coupled and does not signal via Gas in brown adipocytes, the effect of TUG-891 on intracellular cAMP levels was determined. As expected, TUG-891 had no effect on cAMP production (Appendix Fig S9D)."

- Data on protein phosphorylation of downstream cAMP targets *in vivo* have been added to the Results section (page 11, lines 325-327):

"Protein (Appendix Fig S3B-E) and gene (Appendix Fig S3F) expression of markers for lipolysis, adipogenesis, proliferation and thermogenesis were largely unaffected in BAT."

Minor points:

- **Make sure that the recent literature on Gq signaling and GPCRs in BAT is cited correctly.**

We apologize for this oversight. The citations of Gorski *et al.* and Quesada-López *et al.* have now been completed.

- **Hudson et al., 2013, Mol Pharmacol show that TUG-891 is a potent agonist of FFA4 with only limited selectivity for mouse FFA1, complicating its use in vivo. Therefore the authors use GPR120 knockout mice. They state (line252) "In GPR120 KO mice, TUG-891 failed to significantly reduce total body weight (Fig. 3A) and fat mass (Fig 3B),..." The authors should rephrase this, because TUG clearly reduces both body weight and fat mass, albeit not significantly.**

Authors' reply:

We agree with the reviewer that there is a mild effect of TUG-891 on body weight and fat mass in GPR120 KO mice. We have rephrased this section as shown below.

Changes made to the manuscript:

- This part in the Result section has been altered (page 11, lines 333-334):

"In GPR120 KO mice, TUG-891 non-significantly reduced body weight (Fig 3A) and fat mass (Fig 3B), but not to the same extent as in WT mice."

- **TUG treated mice have reduced food intake. However, the authors did not discuss whether increased fat turnover could be centrally induced as well. This should be discussed. A tissue-specific deletion of GPR120 only in BAT would be nice to have, but is clearly beyond the scope of this study.**

Authors' reply:

To our knowledge, it has not been studied whether TUG-891 is able to pass the blood brain barrier. However, carboxylic acids similar to TUG-891 show difficulty penetrating the CNS, making it unlikely for TUG-891 to exert metabolic effects by affecting the brain. Nevertheless, we discuss this possibility in the revised manuscript and propose using tissue-specific GPR120 KO mice for future experiments.

Changes made to the manuscript:

- Information on this topic has been added to the Discussion section (page 17, lines 517-528):

"This indicates that the effects of TUG-891 in vivo could all have been mediated by direct BAT activation. However, we cannot exclude involvement of other tissues, as GPR120 is not exclusively expressed on brown adipocytes. For example, GPR120 is expressed in the hypothalamus and central agonism of GPR120 has been shown to affect energy metabolism (Auguste et al, 2016, Dragano et al, 2017). However, as carboxylic acids similar to TUG-891 have difficulty penetrating the blood brain barrier, this is not very likely (Pajouhesh & Lenz, 2005). ... Future experiments with tissue-specific GPR120 KO mice would be valuable to assess tissue specificity of TUG-891."

Referee #2:

The paper describes targeting of Gpr120 to increase BAT function and thereby improve metabolism. Use of a Gpr120 agonist leads to a protection from diet induced obesity together with an improved metabolism. They furthermore show that activation of Gpr120 leads to increased brown fat adipogenesis in a Ca dependent manner.

The paper is technically well done and presented. It has to be noted that the effect of Gpr120 has already been described by the group of Villaroya, the novelty and clinical translation of this work is based on the use of a Gpr120 agonist which might also be utilized clinically. Therefore, in my opinion it fits very well to the scope of EMBO Molecular Medicine.

There are a few points which need to be addressed in my opinion:

1. The data in Fig.1 needs to be normalized to lean body mass. This might actually change the statement about reduced utilization of glucose.

Authors' reply:

We have now normalized all figures depicting energy expenditure, fat oxidation and glucose oxidation to lean body mass. This normalization did not affect the results of Figure 1, as the mice were housed in metabolic cages during the first week of treatment when lean mass was equal in both groups.

2. The data presented in Fig. 2 on inguinal WAT suggests the presence of more beige cells, is this correct. Did the authors look at gene expression of thermogenic genes? How do the authors envisage that these cells are formed (see point 5) and do the authors expect that this contributes to the overall phenotype. This should be discussed in detail (see also point 4).

Authors' reply:

Indeed, *Ucp1* expression was markedly increased in gWAT of TUG-891-treated mice (Appendix Fig S3H). UCP1 staining of adipose tissues confirmed increased UCP1 levels in gWAT and showed a trend towards increased UCP1 in sWAT (Appendix Fig S4). The increased UCP1 expression in white fat could be due to either transdifferentiation of white adipocytes into beige adipocytes, or proliferation of progenitor cells and their differentiation into beige adipocytes. Increased expression of proliferation markers (i.e. *Ccna*, *Ccnb* and *Mki67*) in white fat of TUG-891-treated mice supports the latter possibility (Appendix Fig S3G-H). These data are consistent with an earlier report on the role of GPR120 in browning of WAT (Quesada-López *et al*, 2016), however, it remains to be determined to what extent this contributes to thermogenesis (Kalinovich *et al*, 2017).

Changes made to the manuscript:

- These data have been added to the Results section (page 11, lines 325-329):

“Protein (Appendix Fig S3B-E) and gene (Appendix Fig S3F) expression of markers for lipolysis, adipogenesis, proliferation and thermogenesis were largely unaffected in BAT. However, Ucp1 gene expression (Appendix Fig S3H) and protein staining (Appendix Fig S4) was increased in gWAT of TUG-891-treated animals, suggesting GPR120-mediated browning.”

3. The food intake is interesting especially since the use of ko mice shows a similar trend in response to treatment (albeit not significant), suggesting that at least a small part of the phenotype is mediated by off-target effects. This should be discussed.

Authors' reply:

Our results indeed show a reduction in food intake in both wildtype and GPR120 KO mice, indicating an effect of TUG-891 that is not mediated through GPR120. TUG-891 also has some affinity for another G protein-coupled receptor, namely GPR40, which has been shown to affect food intake in mice (Gorski *et al*, 2017). Therefore, we explain in the Discussion section of the manuscript that effects of TUG-891 on food intake might be GPR40-mediated (pages 17, lines 505-510):

“GPR40 is involved in insulin secretion and glucose metabolism (El-Azzouny et al, 2014, Itoh et al, 2003), and GPR40 KO mice develop obesity, glucose intolerance and insulin resistance (Kebede et al, 2008). Also, activation of GPR40 has recently been shown to reduce food intake and body weight in mice (Gorski et al, 2017). In our study, food intake was reduced in both wild type and GPR120 KO mice treated with TUG-891. Therefore, this effect might be mediated through GPR40 instead of GPR120.”

4. In general, it is problematic to make the link between BAT activation and the observed phenotype without experiments at thermoneutrality. I am happy that the authors have phrased their manuscript very carefully in that regard, nevertheless this point should be mentioned.

Authors' reply:

We agree with the reviewer that it would be interesting to perform these experiments at thermoneutrality. We believe that this would result in even greater differences between the vehicle- and TUG-891-treated mice, as there is little to no basal BAT activation at thermoneutrality. We have suggested performing experiments at thermoneutrality as future research in the Discussion section.

Changes made to the manuscript:

- The possibility of performing experiments at thermoneutrality has been suggested as future research in the Discussion section (page 17 lines 528-530):

“In addition, it would be interesting to repeat our in vivo experiments at thermoneutrality, to substantiate the link between BAT activation and the observed phenotype.”

5. The role of Gpr120 in adipogenesis is problematic in my opinion. The only data which supports this notion is the Ap2 expression and maybe the fact that there are more brown adipocytes. Based on the presented data the authors should be able to calculate the overall number of brown adipocytes in BAT from their average cell size data and organ weight. Is this number increased? If yes it would strengthen their point on adipogenesis, if not this part should either be removed or discussed very critically.

Authors' reply:

In the manuscript, we have used the word ‘adipogenesis’ interchangeably with ‘differentiation’ to discuss the process of preadipocytes maturing to brown adipocytes (Fig 6). However, this does not apply when you define adipogenesis as the formation of new adipocytes through cell divisions. We have insufficient evidence to support a role of GPR120 in proliferation of brown adipocytes. An increase in proliferation markers (*Ccna*, *Ccnb* and *Mki67*) was observed in WAT of TUG-891-treated mice (Appendix Fig S3G-H), but this was not the case for BAT

(Appendix Fig S3F). Also, TUG-891 treatment reduced iBAT weight and lipid droplet content quite comparably, with -31% and -28% respectively (Fig 2). A larger relative decrease in lipid droplet content compared to relative decreased total iBAT weight would suggest an increased number of brown adipocytes. However, this was not the case. To avoid confusion regarding semantics, we have changed ‘adipogenesis’ in the manuscript to ‘differentiation’.

6. From my point of view, it looks like TUG might mainly regulate BAT activity (based on the outcome of the calculation in point 6). This should be better reflected in the abstract.

Authors’ reply:

We agree that our data suggest that TUG-891 mainly regulates BAT activity. However, as GPR120 is not exclusively expressed in BAT, we cannot exclude the involvement of other tissues in metabolic effects of TUG-891. This point was also raised by the other reviewers. Therefore, we refrain from suggesting that metabolic effects of TUG-891 were all due to increased BAT activity, and now suggest the possibility of using tissue-specific KO mice for future experiments to assess specificity.

Changes made to the manuscript:

- The suggestion of using tissue-specific GPR120 KO mice to evaluate specificity of TUG-891 has been added to the Discussion section (page 17, lines 517-528):

“This indicates that the effects of TUG-891 in vivo could all have been mediated by direct BAT activation. However, we cannot exclude involvement of other tissues, as GPR120 is not exclusively expressed on brown adipocytes. ... Future experiments with tissue-specific GPR120 KO mice would be valuable to assess tissue specificity of TUG-891.”

Referee #3:

The manuscript by Schilperoort and colleagues presents results of investigations of the effects of the GPR120 agonist TUG891 on mouse metabolism and particularly on brown adipose tissue. Significantly similar data were recently reported from Villarroya's laboratory, which agree with many of the observations here. There are two more unique points, one of which is a concern, the other potentially of interest provided the necessary controls confirm an effect.

1) The first point that worries me is the decrease in lean body mass that the authors report but subsequently ignore, since they say that the loss of lean is only 10 %, while that of lipid is much greater. While this is mathematically correct, I think the authors should be considerably more concerned, since most of the lean mass is unlikely to change (bones, brain, kidney etc.). They demonstrate a decrease in liver weight but have apparently not measured skeletal muscle. This would be very important to measure to evaluate potential negative effects of the treatment.

Authors' reply:

Unfortunately, we did not measure the skeletal muscle weight. To assess whether the treatment affected muscle tissue we examined gene expression of various markers of inflammation, fibrosis, atrophy and regeneration. While markers for inflammation and fibrosis were not affected by treatment, markers for both muscle atrophy and regeneration tended to be increased. These results suggest a higher muscle turnover which could be caused by off-target effects of TUG-891. Alternatively, effects on muscle could have been mediated through GPR120, as literature is still divided on whether GPR120 signaling plays a role in muscle cells.

Changes made to the manuscript:

- These data have been added to the Results section (page 10, lines 309-311):

"The reduced lean mass could be due to increased muscle turnover, as TUG-891 non-significantly increased expression of markers for both muscle atrophy and regeneration (Appendix Fig S1)."

- Information on this topic has been added to the Discussion section (page 17, lines 523-527):

"Also, conflicting reports exist on whether GPR120 plays a role in muscle physiology and metabolism (Kim et al, 2015, Oh et al, 2010). In our study, expression of Gadd45a, Murf1 and Myog in skeletal muscle tissue was mildly affected by TUG-891 treatment. Whether this is an off-target effect or GPR120-mediated effect, which could affect muscle function, remains to be investigated."

2) The second point refers to the observation of an acute stimulation of respiration by TUG in isolated adipocytes. This could be an interesting observation but its validity must be checked. There are several possible reasons for this result. The agonist could stimulate increases in cyclic AMP and the respiration would then follow as it does with e.g. norepinephrine. This is probably not very likely. However, was the effect inhibited by AH7614? If so, this would add credibility to it being a somewhat specific effect.

Authors' reply:

To exclude potential involvement of cAMP mediated signaling, we have treated cells with vehicle, TUG-891, or forskolin and measured intracellular cAMP levels. As expected, we found that our positive control forskolin increased cAMP levels, while TUG-891 did not (Appendix Fig S9D). Also, we have pretreated cells with AH7614 to study the specificity of the TUG-891-induced respiration. AH7614 reduced the stimulated respiration by approx-

imately -50% (Fig 7A), indicating that the effect of TUG-891 is both GPR120-dependent and GPR120-independent (see next remark).

Changes made to the manuscript:

- Data involving cAMP have been added to the Results section (page 14, lines 430-432):

“To ensure that GPR120 is Gαq coupled and does not signal via Gas in brown adipocytes, the effect of TUG-891 on intracellular cAMP levels was determined. As expected, TUG-891 had no effect on cAMP production (Appendix Fig S9D).”

- Data from the new AH7614 experiment have been added to the Results section (page 14, lines 410-413):

“Strikingly, TUG-891 acutely increased the O₂ consumption rate of brown adipocytes by more than twofold (Fig 7A). Pretreatment with the GPR120 antagonist AH7614 reduced rather than abolished this response (Fig 7A), indicating that TUG-891 exhibits both GPR120-dependent and GPR120-independent activity.”

The agonist could be a direct activator of UCP1, in a similar manner to e.g. fatty acids and retinoic acid. This would be interesting and potentially of therapeutic significance. However, to investigate this further, the authors need to convince me that the effect is not an artifact of the system. Thus, they should repeat the experiments under conditions as indicated in e.g. Li et al. EMBO Rep. 2014 in the presence of 2 % BSA. It would also help if UCP1 KO cells could be used to demonstrate some specificity. Essentially all carboxylic acids can uncouple any mitochondria given sufficient concentration!

Authors' reply:

The reviewer makes a very good point. We have now measured TUG-891-induced respiration in isolated mitochondria from wild type and UCP1-deficient mice. Interestingly, at concentrations ranging from 10-40 μM (10 μM TUG-891 has been used in all *in vitro* experiments), TUG-891 increased respiration in wild type but not UCP1-deficient mitochondria (Fig 7B&C). These results indicate that TUG-891 indeed activates UCP1 directly, presumably without involvement of GPR120 signaling. This also explains the partial but not full inhibition of respiration following AH7614 pretreatment, and suggests mechanisms of action both dependent and independent of GPR120.

Changes made to the manuscript:

- This data has been added to the Results section (page 14, lines 414-427):

“We investigated whether TUG-891 functions in a manner similar to LCFAs which can directly activate UCP1 by measuring O₂ consumption in isolated BAT mitochondria in conditions mimicking a cellular environment with high purine nucleotide (GDP) content and inhibited UCP1 (Matthias et al, 2000). Indeed, TUG-891 (≥ 10 μM) increased O₂ consumption in mitochondria isolated from WT mice (Fig 7B), suggesting that TUG-891 has the capacity to overcome purine nucleotide inhibition and activate UCP1 in brown adipocytes. TUG-891 also increased O₂ consumption in mitochondria from UCP1 KO mice, but this effect was smaller and occurred at higher concentrations (≥ 90 μM) as compared to WT mitochondria (Fig 7C), a response that is also observed with oleate (Shabalina et al, 2004). As oxidative capacity (FCCP response, Fig 7B&C) of WT and UCP1 KO mitochondria was equal, these results suggest that TUG-891 increases mitochondrial respiration through activation of UCP1 (Appendix Fig S9A). TUG-891 exhibited a competitive interaction with GDP in WT but not UCP1 KO mitochondria (Appendix Fig S9B&C), further supporting the effect of TUG-891 on UCP1.”

- Information on this topic has been added to the Discussion section (pages 17-18, lines 531-540):

“Using a GPR120 antagonist, we discovered that the TUG-891-induced increase in O₂ consumption in brown adipocytes is only partly mediated by GPR120. The GPR120-independent effect of TUG-891 could be due to direct activation of mitochondrial UCP1. TUG-891 relieves the natural inhibition of UCP1 by GDP (Matthias et al, 2000), similar to oleate and other LCFAs (Fedorenko et al, 2012, Shabalina et al, 2004), thereby leading to increased UCP1 activity and uncoupled mitochondrial respiration. This could explain the moderately decreased fat mass and increased FA uptake by BAT in TUG-891-treated GPR120 KO mice. However, as most metabolic effects of TUG-891 were largely attenuated or abolished in GPR120 KO mice, BAT activation by TUG-891 in vivo is mainly dependent on GPR120 signaling.”

Other points:

3) The observation of an increase in intracellular calcium following TUG treatment in isolated adipocytes is an entirely expected response to a Gq-coupled agonist, similarly to the response to an alpha1-adrenergic agonist. Note however that negative effects of BAPTA do not show that the increased oxygen consumption is stimulated through calcium, merely that cytosolic and mitochondrial calcium homeostasis is necessary for most responses in cells and that matrix calcium is required for several citric acid cycle dehydrogenases.

Authors' reply:

Indeed, we cannot be sure whether calcium availability is necessary for the response to TUG-891, or whether calcium is directly mediating this response. Therefore, we have formulated our results more careful in this regard. Also, to substantiate the role of calcium in GPR120-mediated activation of brown adipocytes, we have further examined the potential mechanism through which the induction of calcium could increase respiration. Experiments using both membrane potential dependent and independent MitoTracker dyes suggest the involvement of calcium-induced mitochondrial depolarization, and subsequently fragmentation. We elaborate on this potential mechanism in the revised manuscript.

Changes made to the manuscript:

- New mechanistic data has been added to the Results section (page 15, lines 442-453):

“As Ca²⁺ could affect mitochondrial polarization, effects of TUG-891 on mitochondrial membrane potential was investigated. Cells were incubated with MitoTracker Green FM (MTG) and MitoTracker Red CMXRos (MTR), which stain mitochondria independent of and dependent on membrane potential, respectively. Relative intensity (MTR/MTG) of these stainings can be used as a measure for mitochondrial polarization. Stimulation with TUG-891 resulted in fading of the MTR signal while the MTG signal remained intense, indicative of mitochondrial depolarization (Fig 7F). In addition, mitochondria were more fragmented following TUG-891 stimulation (Fig 7G), pointing towards increased mitochondrial fission, which could explain the GPR120-dependent increase in respiration. Of note, the timing of TUG-891-induced changes in mitochondrial morphology coincide with increases in intracellular Ca²⁺, suggesting this effect is mediated through Ca²⁺.”

Changes made to the manuscript:

- Information on this topic has been added to the Discussion section (page 18, lines 545-559):

“A recent study showed that Ca²⁺ could increase respiration in brown adipocytes by decreasing the mitochondrial membrane potential (MMP) (Hou et al, 2017). Evidently, the β receptor agonist isoprenaline induces Ca²⁺ release from the endoplasmic reticulum of brown adipocytes resulting in mitochondrial depolarization and fission (Hou et al, 2017), the latter being a process required for NA-induced uncoupled respiration (Wikstrom et al, 2014). Therefore, we studied whether this Ca²⁺-mediated pathway of mitochondrial depolarization and fission could also underlie GPR120-mediated activation of brown adipocytes. Mitochondria were co-stained with the MMP sensitive MitoTracker CMXRos (MTR) and the MMP insensitive MitoTracker Green (MTG) (Pendergrass et al, 2004), and

the relative ratio of MTR/MTG was used as a measure for mitochondrial depolarization (as seen in (Wikstrom et al, 2014) in which TMRE was used instead of MTR). TUG-891 stimulation resulted in a reduction in the MTR/MTG ratio, indicative of mitochondrial depolarization. Also, TUG-891 increased mitochondrial fragmentation, presumably secondary to Ca²⁺-induced mitochondrial depolarization. These results suggest that GPR120 signaling could increase metabolic activity of brown adipocytes by stimulation of mitochondrial fission in a Ca²⁺-dependent manner."

4) The authors note on p. 14 that "little is known about the effects of Ca in brown adipocytes"; this is hardly a scientifically valid statement. If the authors put in "brown fat" and "calcium" in PubMed, they will obtain approximately 250 references, many of which, starting in the earlier 1970s through to the late 1990s, investigate a role of calcium in brown fat mitochondria and isolated adipocytes!

We have removed the above-mentioned statement from the revised manuscript.

5) There is also something of a conundrum regarding energy expenditure: the wildtype mice lose body weight and decrease food intake without a change in energy expenditure, while the KO mice also decrease food intake but do not decrease body weight. Any comment?

Indeed, using our automated metabolic cage system we did not observe a difference in energy expenditure between TUG-891 treated wild type and GPR120 KO mice. This would suggest the difference in body weight and fat mass between the animals is entirely due to a difference in food intake. However, five days of treatment with TUG-891 did significantly reduce fat mass without affecting food intake (Figure 1), indicating increased energy expenditure. This difference in energy expenditure was not detected using our metabolic cage system, most likely due to a lack in sensitivity.

6) Regarding Fig. 4, the authors should show the uptake per organ/depot since the decrease in weight is presumably only triglyceride so that the active components are still present, such that even an unchanged actual uptake per depot would appear to be an increase given per g tissue.

We have now also included the uptake per organ in the manuscript in addition to the uptake per gram tissue (Appendix Fig S6). This figure does not include data for sWAT, sBAT and skeletal muscle, as these organs cannot be taken out quantitatively within an acceptable time frame. As can be appreciated from the new figure, FA uptake per whole iBAT remains significantly increased, consistent with increased BAT activity.

Changes made to the manuscript:

- These data have been added to the Results section (page 12, lines 349-352):

"However, when uptake data were corrected for organ weight (for organs that could be removed quantitatively within an acceptable time frame), this difference in glucose uptake was lost. FA uptake in whole iBAT remained approximately twice as high in treated WT mice versus controls (Appendix Fig S6), showing an independency of organ weight."

7) The reference to Quesada-López is incomplete.

Authors' reply:

We apologize for this oversight. The reference has now been completed.

References

- Auguste S, Fiset A, Fernandes MF, Hryhorczuk C, Poitout V, Alquier T, Fulton S (2016) Central Agonism of GPR120 Acutely Inhibits Food Intake and Food Reward and Chronically Suppresses Anxiety-Like Behavior in Mice. *Int J Neuropsychopharmacol* 19
- Dragano NRV, Solon C, Ramalho AF, de Moura RF, Razolli DS, Christiansen E, Azevedo C, Ulven T, Velloso LA (2017) Polyunsaturated fatty acid receptors, GPR40 and GPR120, are expressed in the hypothalamus and control energy homeostasis and inflammation. *J Neuroinflammation* 14: 91
- El-Azzouny M, Evans CR, Treutelaar MK, Kennedy RT, Burant CF (2014) Increased glucose metabolism and glycerolipid formation by fatty acids and GPR40 receptor signaling underlies the fatty acid potentiation of insulin secretion. *J Biol Chem* 289: 13575-88
- Fedorenko A, Lishko PV, Kirichok Y (2012) Mechanism of fatty-acid-dependent UCP1 uncoupling in brown fat mitochondria. *Cell* 151: 400-13
- Gorski JN, Pachanski MJ, Mane J, Plummer CW, Souza S, Thomas-Fowlkes BS, Ogawa AM, Weinglass AB, Di Salvo J, Cheewatrakoolpong B, Howard AD, Colletti SL, Trujillo ME (2017) GPR40 reduces food intake and body weight through GLP-1. *Am J Physiol Endocrinol Metab* 313: E37-E47
- Gotoh C, Hong YH, Iga T, Hishikawa D, Suzuki Y, Song SH, Choi KC, Adachi T, Hirasawa A, Tsujimoto G, Sasaki S, Roh SG (2007) The regulation of adipogenesis through GPR120. *Biochem Biophys Res Commun* 354: 591-7
- Hou Y, Kitaguchi T, Kriszt R, Tseng YH, Raghunath M, Suzuki M (2017) Ca²⁺-associated triphasic pH changes in mitochondria during brown adipocyte activation. *Mol Metab* 6: 797-808
- Itoh Y, Kawamata Y, Harada M, Kobayashi M, Fujii R, Fukusumi S, Ogi K, Hosoya M, Tanaka Y, Uejima H, Tanaka H, Maruyama M, Satoh R, Okubo S, Kizawa H, Komatsu H, Matsumura F, Noguchi Y, Shinohara T, Hinuma S et al. (2003) Free fatty acids regulate insulin secretion from pancreatic beta cells through GPR40. *Nature* 422: 173-6
- Kalinovich AV, de Jong JM, Cannon B, Nedergaard J (2017) UCP1 in adipose tissues: two steps to full browning. *Biochimie* 134: 127-137
- Kebede M, Alquier T, Latour MG, Semache M, Tremblay C, Poitout V (2008) The fatty acid receptor GPR40 plays a role in insulin secretion in vivo after high-fat feeding. *Diabetes* 57: 2432-7
- Kim N, Lee JO, Lee HJ, Kim HI, Kim JK, Lee YW, Lee SK, Kim SJ, Park SH, Kim HS (2015) Endogenous Ligand for GPR120, Docosahexaenoic Acid, Exerts Benign Metabolic Effects on the Skeletal Muscles via AMP-activated Protein Kinase Pathway. *J Biol Chem* 290: 20438-47
- Klepac K, Kilic A, Gnad T, Brown LM, Herrmann B, Wilderman A, Balkow A, Glode A, Simon K, Lidell ME, Betz MJ, Enerback S, Wess J, Freichel M, Bluher M, Konig G, Kostenis E, Insel PA, Pfeifer A (2016) The Gq signalling pathway inhibits brown and beige adipose tissue. *Nat Commun* 7: 10895
- Liu D, Wang L, Meng Q, Kuang H, Liu X (2012) G-protein coupled receptor 120 is involved in glucose metabolism in fat cells. *Cell Mol Biol (Noisy-le-grand)* 58: 1757-62
- Matthias A, Ohlson KB, Fredriksson JM, Jacobsson A, Nedergaard J, Cannon B (2000) Thermogenic responses in brown fat cells are fully UCP1-dependent. UCP2 or UCP3 do not substitute for UCP1 in adrenergically or fatty acid-induced thermogenesis. *J Biol Chem* 275: 25073-81
- Oh DY, Talukdar S, Bae EJ, Imamura T, Morinaga H, Fan W, Li P, Lu WJ, Watkins SM, Olefsky JM (2010) GPR120 is an omega-3 fatty acid receptor mediating potent anti-inflammatory and insulin-sensitizing effects. *Cell* 142: 687-98
- Ootsuka Y, Blessing WW (2006) Thermogenesis in brown adipose tissue: increase by 5-HT_{2A} receptor activation and decrease by 5-HT_{1A} receptor activation in conscious rats. *Neurosci Lett* 395: 170-4
- Pajouhesh H, Lenz GR (2005) Medicinal chemical properties of successful central nervous system drugs. *NeuroRx* 2: 541-53

- Pendergrass W, Wolf N, Poot M (2004) Efficacy of MitoTracker Green and CMXRosamine to measure changes in mitochondrial membrane potentials in living cells and tissues. *Cytometry A* 61: 162-9
- Quesada-López T, Cereijo R, Turatsinze JV, Planavila A, Cairó M, Gavaldà-Navarro A, Peyrou M, Moure R, Iglesias R, Giralt M, Eizirik DL, Villarroya F (2016) The lipid sensor GPR120 promotes brown fat activation and FGF21 release from adipocytes. *Nat Commun* 7: 13479
- Shabalina IG, Jacobsson A, Cannon B, Nedergaard J (2004) Native UCP1 displays simple competitive kinetics between the regulators purine nucleotides and fatty acids. *J Biol Chem* 279: 38236-48
- Wikstrom JD, Mahdavian K, Liesa M, Sereda SB, Si Y, Las G, Twig G, Petrovic N, Zingaretti C, Graham A, Cinti S, Corkey BE, Cannon B, Nedergaard J, Shirihai OS (2014) Hormone-induced mitochondrial fission is utilized by brown adipocytes as an amplification pathway for energy expenditure. *Embo j* 33: 418-36
- Zhao J, Cannon B, Nedergaard J (1997) alpha1-Adrenergic stimulation potentiates the thermogenic action of beta3-adrenoreceptor-generated cAMP in brown fat cells. *J Biol Chem* 272: 32847-56

Thank you for the submission of your revised manuscript to EMBO Molecular Medicine. We have now received the enclosed report from the referee who was asked to re-assess it. As you will see, reviewer 1 remains supportive while still requesting additional data. Therefore, I would like to encourage you to address the following:

1) Please address referee 1's comments about GPR120 signalling, experimentally in BAT, and make sure to discuss the findings appropriately, citing correct references.

***** Reviewer's comments *****

Referee #1 (Remarks for Author):

The authors need to show that GPR120 is coupled to Gq in brown adipocytes. Previous studies have shown that activation of Gq in brown adipocytes inhibits differentiation. Moreover, the authors now state "These results indicate that TUG-891 indeed activates UCP1 directly, presumably without involvement of GPR120 signaling." This is a crucial point to strengthen the major findings of this manuscript.

The discussion on the role of Gq in brown adipocytes should be more balanced.

(begins on next page)

Detailed response to the editors' and reviewers' comments

1) Please address referee 1's comments about GPR120 signalling, experimentally in BAT, and make sure to discuss the findings appropriately, citing correct references. Please provide a letter INCLUDING the reviewer's reports and your detailed responses to their comments (as Word file).

Referee #1 (Remarks for Author):

The authors need to show that GPR120 is coupled to Gq in brown adipocytes. Previous studies have shown that activation of Gq in brown adipocytes inhibits differentiation. Moreover, the authors now state "These results indicate that TUG-891 indeed activates UCP1 directly, presumably without involvement of GPR120 signaling." This is a crucial point to strengthen the major findings of this manuscript. The discussion on the role of Gq in brown adipocytes should be more balanced.

Authors' reply:

Indeed, our results suggest that TUG-891 directly activates UCP1, which could contribute to beneficial metabolic effects of the compound. However, the fact that effects of TUG-891 on body weight, fat mass and fat oxidation *in vivo* were largely reduced or absent in GPR120 KO mice indicates that GPR120 signaling is important for these therapeutic effects of TUG-891. To substantiate involvement of Gαq signaling, we have now examined the GPR120-induced Ca²⁺ release in brown adipocytes *in vitro* following incubation with the Gαq inhibitor YM-254890. Stimulation with TUG-891 did not induce Ca²⁺ release in adipocytes pretreated with YM-254890 (Appendix Fig S9H), confirming dependency on Gαq signaling.

Changes made to the manuscript:

- These data have been added to the Results section (pages 8-9, lines 244-246):

'The Gαq inhibitor YM-254890 also blocked the Ca²⁺ response (Appendix Fig S9H), indicating that this effect of GPR120 activation is indeed mediated via Gαq signaling.'

Corresponding Authors: Maaïke Schilperoort and Mark Christian

Manuscript Number: EMM-2017-08047